



# Experimental study of sediment traps permeable for frequent floods

Sebastian Schwindt[1,2], Mário J. Franca[1,3], Alessandro Reffo[4], Anton J. Schleiss[1]

[1]Laboratory of Hydraulic Constructions (LCH), École polytechnique fédérale de Lausanne (EPFL), Lausanne, CH-1015, Lausanne, Switzerland
5  [2]Department of Land, Air and Water Resources, UC Davis, Davis, 95616, California, USA
[3]River Basin Development, IHE Delft Institute for Water Education, Delft, NL-2611, The Netherlands
[4]University of Trento, Trento, I-38050, Italy

*Correspondence to*: Sebastian Schwindt (sschwindt@ucdavis.edu)

10  **Abstract.** Sediment traps created by partially open torrential barriers are crucial elements for flood protection in alpine regions. The trapping of sediment is necessary when intense sediment transport occurs during floods that may endanger urban areas at downstream river reaches. In turn, the unwanted permanent trapping of sediment during small, non-hazardous floods can result in the ecological and morphological depletion of downstream reaches. This study experimentally analyses a new concept for permeable sediment traps. For ensuring the sediment transfer up to small floods, a guiding channel implemented in the deposition area of the sediment trap was studied systematically. The bankfull discharge of the guiding channel corresponds to a dominant morphological discharge. At the downstream end of the guiding channel, a permeable barrier triggers sediment retention and deposition. The permeable barrier consists of a bar screen for mechanical deposition control, superposed to a flow constriction for the hydraulic control. The fail-safe clogging of the barrier and the sediment deposition upstream can be ensured for discharges that are higher than the bankfull discharge of the guiding channel.

## 1 Introduction

The sediment supply of mountain rivers is a substantial source for the dynamics of river ecosystems. Artificial barriers, such as dams, can affect the natural flow regime variability with direct impacts on the eco–morphological state of rivers (Allan and Castillo, 2007; Sponseller et al., 2013). Maintaining the natural conditions of rivers is a multidisciplinary concern and artificial interventions require the consideration of ecological and morphological site evaluations (Bain et al., 1999).

25  The morphological processes in mountain rivers depend and interact with the transport of sediment (Buffington and Montgomery, 1999; Hassan et al., 2005; Recking et al., 2016). The sediment supplied by the headwaters is also essential for the ecologic diversity of downstream river reaches (Denic and Geist, 2015; Gomi et al., 2002; Milhous, 1998). Therefore, sediment transport-related criteria can also be designated as "eco–morphological" river characteristics (Moyle and Mount, 2007). These characteristics can often be attributed to a certain discharge which alters and rearranges the channel bed

30  morphology, which may be assessed by morphologically effective (dominant) discharge (Wolman and Leopold, 1957a, 1957b; Wolman and Miller, 1960).





For estimating the transport capacity of the headwaters, many (semi-) empirical formulae were developed (Meyer-Peter and Müller, 1948; Recking, 2013; Smart and Jaeggi, 1983; Wilcock, 2008). However, the sediment transport in such streams is often driven by the sediment supply from bed-external sources, as long as armour breaking does not occur. In such co- or non-alluvial channels, the fluvial sediment transport can be assessed in terms of the finer "traveling bed load" (Piton and Recking, 2017; Yu et al., 2009). The characteristic grain size of the traveling bed load can be estimated by the grain size of sediment bars along the channel banks upstream. These bars are silent witnesses of earlier flood events and contain information about sediment transport during past floods (Kaitna and Hübl, 2013). The application of the grain size of the traveling bed load to bed load transport formulae can be used for establishing sediment rating curves, as a computation basis for the dominant discharge.

The sediment transport in headwaters can be disturbed by hydraulic structures for water use as, for instance, hydropower, drinking water, or flood protection (Kondolf, 1997; Lane et al., 2014; Williams and Wolman, 1984). This may cause sediment deficits in downstream reaches resulting in bed incision as well as the erosion of channel banks and floodplains (Pasternack and Wyrick, 2017).

The artificially forced retention of sediment, especially bed load, may be required for exceptional floods that endanger potentially downstream riparian urban areas. This purpose can be achieved by the construction of sediment traps, upstream of endangered areas (e.g., Hampel, 1968; Hübl et al., 2005; Kronfellner-Krauss, 1972; Mizuyama, 2008; Piton and Recking, 2016 a; Wang, 1903). However, history shows that such artificial interventions contribute to the eco-morphological depletion of mountain rivers (Comiti, 2012).

Bearing in mind the above mentioned aspects of assessing sediment transport and the necessity of sediment transport continuity in mountain rivers, this study revises the design of sediment traps with open (or permeable) check dams. New design elements are conceived to fit the real needs of land protection with limited impacts on the river ecology in terms of the continuity of sediment transport. To achieve this objective, this experimental and praxis-oriented research introduces a guiding channel across the retention reservoir of a sediment trap as a new design element and a combination of two barriers types for improving the functionality of open check dams is proposed.

## 2 Design approach for permeable sediment traps

The typical concept of sediment traps is shown in Figure 1, with the following elements: (1) a barrier with an opening (open check dam) having an open or close crest and (2) downstream abutments with a counter sill for scour protection; (3) a retention basin, i.e., deposition area; (4) lateral dykes for limiting the deposition area; (5) a maintenance access; and (6) an inlet sill with scour protection.

The river discharge should pass the deposition area and the barrier opening(s) without interaction, unless intense bed load transport occurs. The triggering of bed load retention can be a result of hydraulic control since a certain flood discharge is exceeded or mechanical control due to entangled coarse sediment or wood.





Previous studies have shown that the retention of bed load is hydraulically initiated as soon as the barrier causes a hydraulic jump upstream underlying generally supercritical flow conditions (Schwindt et al., 2017a, 2017b). Supercritical flow conditions are typical in mountain rivers but not omnipresent (e.g., Heyman et al., 2016). This study is limited to supercritical flow conditions, where the Froude number in the non-constricted channel is generally larger than unity. In this case, an opening

in the barrier acts like a vertical or lateral flow constriction that causes backwater in the deposition area during floods. Therefore, the free surface flow capacity of the barrier opening(s) without backwater should be smaller than that of the sediment-laden flood discharge that potentially endangers urbanized downstream regions.

The mechanically caused sediment retention occurs when the size of the transported sediment is too large to pass the openings. In practice, the mechanical sediment retention is typically achieved by screen or net structures (Piton and Recking, 2016a).

The combination of both mechanical and hydraulic control mechanisms can be obtained by installing a bar screen in front of an opening of a barrier (open check dam). This combination has been shown to be advantageous to avoid the unwanted flushing of formerly deposited sediment in the deposition area (Schwindt et al., 2017). The implementation of a bar screen for the mechanical control and a flow constriction for the hydraulic control (B in Fig. 1) is experimentally and systematically analysed here. The design of the bar screen refers to criteria from the literature (Piton and Recking, 2016a; Uchiogi et al., 1996;

Watanabe et al., 1980).

In the context of river continuity, inlet structures (Fig. 1) in the form of sills are, besides the barrier itself, an additional obstacle regarding the longitudinal river connectivity. Such sills can cause downstream scour or dead storage volume (Zollinger, 1983). Therefore, inlet structures are avoided when possible in practice (Piton and Recking, 2016a), and consequently, they are not considered in the present study.

As an important novel feature, a guiding channel (A in Fig. 1) across the deposition area is subsequently introduced. The guiding channel has the purpose of ensuring sediment transfer up to its bankfull discharge. The experiments refer to a theoretical bankfull discharge of the guiding channel that corresponds to the dominant, morphologically effective discharge. The use and performance of the guiding channel is experimentally considered in combination with barriers for the hydraulic and mechanical deposition controls (B in Fig. 1 and Schwindt et al., 2017a).

The new concept of a permeable sediment trap (Fig. 1) is tested with a standardized hydrograph, corresponding to typical hydrological characteristics of mountain rivers. Special attention is drawn in supplementary experimental runs on the possibility of flushing of sediments.

## 3. Sediment deposition processes and controls

Independent of the deposition control mechanism, the barrier can cause backwater, where sediment deposition occurs due to

the deceleration of the flow with a consequent reduction in the energy slope (Armanini et al., 1991; Armanini and Larcher, 2001; Leys, 1976; Mizuyama, 2008; Zollinger, 1984). However, the patterns of sediment deposits in the deposition area differ for both obstruction mechanisms, as shown in Figure 2 (Lange and Bezzola, 2006; Piton and Recking, 2016a). In the case of





hydraulic control, the bed load settles in the backwater, immediately downstream of the hydraulic jump, and forms a delta-like deposit. For coarse bed load, the deposit evolves in the upstream direction; for fine bed load, the deposit evolves in the downstream direction (Armanini and Larcher, 2001; Campisano et al., 2014; Jordan et al., 2003). The mechanical clogging of the barrier causes a hydraulic jump immediately upstream of the barrier. Thus, the formation of the sediment deposit is initiated

directly upstream of the barrier and displaces the hydraulic jump in the upstream direction. This results in the successive formation of an elongated sediment deposit that evolves in the upstream direction until it reaches the level of the barrier crest (flood control). Then, a second deposit layer forms on top of the former. This layer-wise deposition continues in a succession of quasi-equilibrium states until the deposition area is completely filled up to the barrier crest (Campisano et al., 2014; Piton and Recking, 2016a).

Low clearance heights or narrow clearance widths of the opening(s) in the barrier may interrupt the river connectivity with negative effects on the downstream eco–morphological river state (Brandt, 2000; Castillo et al., 2014; Kondolf, 1997). The functioning of barriers for both hydraulic and mechanical retention controls is experimentally examined in the present study. Therefore, the experimental set-up (as used by the authors in Schwindt et al., 2017a; 2017b) included a widened deposition area, according to typical sediment traps in a mountain-river-like environment.

## 4. Methodology

### 4.1 Experimental set-up

The design of the experimental set-up (Figure 3) was inspired by 132 characteristic datasets from mountain rivers (Schwindt, 2017). Thus, even though any particular prototype underlay the model, a geometric scale in the range of 1:10 to 1:40 can be supposed.

The experimental set-up consisted of a sediment supply system, with a container (element 1 in Fig. 3) for the sediment storage and supply rate control by a cylindrical bottom screw, as well as a system of conveyor belts (element 2 in Fig. 3). The sediment supply mixture consisted in fine and medium gravel, characterized by $D_{16} = 6.7$ mm, $D_m = 10.4$ mm, $D_{84} = 13.7$ mm and $D_{max} = 14.8$ mm, in line with the field data. The water was supplied by the laboratory pump system and mixed with the sediment in a 2.5-m-long adaptation reach (element 3 in Fig. 3), which was situated upstream of a 3.0-m-long observation reach

(element 4 in Fig. 3). The minimum and maximum pump discharges were 5.5 l/s and 12.5 l/s, respectively. The barriers (element 5 in Fig. 3) in terms of a bar screen and mobile PVC elements were introduced in the lower third of the observation reach, approximately 0.9 m upstream of the model outlet. A filter basket (element 6 in Fig. 3) at the model outlet served for the separation of outflowing sediments and water. The water returned to the laboratory pump circuit.

The pump discharge was registered second-wise by an electromagnetic flow meter (type ABB FXE4000) with a precision of

0.1 %. The wet outflowing sediments (bed load outflow $Q_{b,o}$) were weighed minute-wise in an intermediate sieve in the filter basket, outside of the flow, by a scale with a precision of ±2 g (type Kern 440 51N). The total weight of the sediment deposits



was measured by an industrial scale (type Dynafor MWXL-5, precision of ±0.01 kg) attached to the filter basket, after the flushing of the sediment deposits, for every test.

The volumes and patterns of the sediment deposits were recorded using a motion-sensing camera (Microsoft Kinect V2) at the end of every test. This application has been shown promising, but the results were still affected by uncertainties (Lachat et al., 2015). For this reason, complementary and redundant reference measurements were made using a laser (type Leica DISTO D410, precision of ±1 mm). Thus, a redundant bathymetric record was produced by centimetre-wise measurements along 16 cross-sections with an interspace of 0.10 m (according to the gridlines indicated in Figure 4), which corresponds to approximately 650 point measurements. The accuracy of both measurement techniques was evaluated using the total weight and the apparent packed density $\rho_s'$ of the sediment (gravel with $\rho_s' = 1\,550$ kg/m³, supplier information). The evolution of the deposit pattern during the hydrograph experiments was observed by a camera (GoPro Hero Silver, 2016) taking top-view time-lapse pictures, every 10 s.

## 4.2 Deposition area with guiding channel

The observation reach included the deposition area with guiding channel and downstream deposition control barrier, according to the sediment trap concept shown in Fig. 1. The geometry of the deposition area referred to the desirable optimum between sediment retention and flushing: the trapping efficiency of reservoirs (Brown, 1943), as well as the sediment flushing potential, which increases with increasing length and decreasing width of the deposition area (Piton and Recking, 2016a; Zollinger, 1983, 1984). The unwanted flushing of sediment traps represents a high risk at urban downstream reaches and should be avoided (Morris et al., 2008; Sodnik et al., 2015). For ensuring a high trapping efficiency, but at the same time limiting the risk of unwanted sediment flushing, a rectangular deposition area with a width to length ratio of 3:4 was used for the experiments (Zollinger, 1983). The opening angle of the deposition area was set to 30°, which is oriented at the opening angle of natural alluvial deposition cones formed by continuous sediment supply (Parker et al., 1998). The here applied barriers resulted from previous experimental analysis (Schwindt et al., 2017a) with a longitudinal channel slope of $S_o = 5.5$ % which can be typically found in co- or non-alluvial mountain rivers (Piton and Recking, 2017; Rosgen, 1994; Yu et al., 2009).

According to the above-mentioned criteria, the deposition area (Fig. 4 a) had a length of 1.60 m, a width of 1.20 m, a longitudinal slope $S_o$ of 5.5 % and an opening angle of 30°. For the description of sediment deposits, a model coordinate system was defined with the origin at the location of the barrier. Thus, the positive x-axis points in the upstream direction and $x = 0$ corresponds to the insertion point of the barrier; the positive y-axis points toward the right bank and $y = 0$ corresponds to the flume centre; the positive z-axis points upward and $z = 0$ corresponds to the flume bottom at the barrier.

The bottom of the deposition area consisted of gravel from the supply mixture. For ensuring the same initial conditions for every experimental run, cement grout was poured over the shaped, loose foundation gravel (cf. Fig. 4 b and c).

The design criteria for torrential barriers regarding the discharge capacity and the effects on bed load transport have been derived in previous studies from flume observations with constraint, monotone channel morphology (Armanini and Larcher, 2001; Piton and Recking, 2016a, 2016b; Schwindt et al., 2017a). The guiding channel (A in Fig. 4) enables not only the




sediment transfer during low flows, but it also ensures the desired hydraulic functioning of the barrier, as it represents a morphological fixation of the monotone channel in deposition area up to the bankfull discharge. In the experiments, the hydraulic design and bankfull discharge of the guiding channel corresponded to "small" discharges, equivalent to the dominant, morphologically effective discharge referring to pristine downstream reaches in practice. A flood hydrograph with higher

discharges than the bankfull discharge of the guiding channel was simulated. Due to the model limitations, the guiding channel had a bankfull discharge of $Q_{bf}$ = 5.5 l/s. In practice, the bankfull discharge should be slightly larger than the effective discharge related to bankfull discharge in order to enhance the eco-morphological flow continuum through the sediment trap.

The guiding channel had a trapezoidal cross-section, as shown in Figure 5, with a bank inclination of $m$ = 2.25 (dimensionless) and a bottom channel width of $w$ = 0.11 m (Schwindt et al., 2017a; 2017b). The channel had a roughness according to a

Mannings' $n$ of $n \approx 0.02$. Grains larger than the $D_{84}$ of the sediment supply mixture constituted the roughness. According to the Gauckler-Manning-Strickler formula, the bankfull discharge of 5.5 l/s corresponds to a normal flow depth of 0.032 m. Also the bed shape of the guiding channel was fixed by pouring cement grout into the voids of the loose grains.

### 4.3 Tested deposition control modes of the barrier

The torrential barrier was introduced at the downstream end of the deposition area (element 5 in Fig. 3 and Schwindt et al.,

2017b). The barrier incorporated a flow constriction for the hydraulic control and a bar screen for the mechanical control of bed load retention. Three cases of deposition control types were considered:

- Case 1 – hydraulic deposition control only, where two situations are considered:

    - *Hy-no-* a non-overflown, infinitely high barrier with constant opening dimensions (Figure 6 a);

    - Hy-o - an overflown barrier with limited height and constant opening height (Fig. 6 b);

- Case 2 – Mec mechanical deposition control by a bar screen with constant spacing (Fig. 6 c); and

- Case 3 – HyMec combined deposition control, i.e., a bar screen upstream of an overflown hydraulic control barrier with variable opening height (Fig. 6 d).

For the hydraulic control only, two types of flow situations were considered (Table 1): case *Hy-no*, with infinite barrier height, where barrier overflow was not possible and case *Hy-o*, with overflown barrier, where the barrier height was limited to 0.11 m.

In case *Hy-no*, the constriction height $a$ was 0.152 m and the constriction width $b$ was 0.076 m. The opening height of 0.152 m corresponded to the technically maximum possible constriction height due to the model limitations; the corresponding width of 0.076 m was required to hydraulically trigger sediment retention when the bankfull discharge of 5.5 l/s was exceed, according to previous studies (Schwindt et al., 2017a). Smaller widths were not considerable to ensure flow and sediment continuity in practice.

The unwanted flushing of sediment has been observed in previous studies when barriers were overflown (e.g., Schwindt et al., 2017a; Zeller, 1973), as considered by the cases *Hy-o*, *Mec* and *HyMec* with limited the barrier height. However, the creation



of a sediment deposit that can be flushed requires the initial impounding without barrier overflow. Thus, the barrier height was determined in a manner that the opening was pressurized for discharges higher than 5.5 l/s and so that the barrier could not be overflown for discharges up to 7.0 l/s corresponding to the first incremental increase of the hydrograph. Barrier overflow can be avoided when the cross-section-averaged energy head is not higher than the barrier (Piton and Recking, 2016a). In the

experimental set-up, the head corresponding to a discharge of 7.0 l/s was approximately 0.11 m, which was decisive for limiting also the barrier height to 0.11 m.

The width of the opening in the overflown hydraulic control barrier (cases *Hy-o* and *HyMec*) was 0.15 m, which was slightly larger than the bottom width of the guiding channel. This choice has been made to minimize the effects of the barrier on the flow when the guiding channel was not overtopped. An opening height of 0.040 m is required to hydraulically trigger sediment

retention for a discharge of 5.5 l/s (Schwindt et al., 2017a; Schwindt et al., 2017b).

For the combined control barrier, also larger opening heights were analysed to study its effect on the deposition control by combined barriers. Thus, the opening heights tested in the case *HyMec* were 0.040 m, 0.043 m and 0.047 m, where the constriction width was kept constant with 0.15 m.

Pure mechanical deposition control (case *Mec*) was tested by a bar screen with a height of 0.11 m and a bar width, as well as

an interspace between the bars corresponding to the $D_{84}$ of the sediment supply mixture. The clearance between the guiding channel bottom and the lower end of the bars was $1.75 \cdot D_{84}$ to ensure the sediment transfer during small discharges and at the same time fail-safe clogging when sediment retention was wanted (Schwindt et al., 2017a). Fail-safe clogging refers to the barrier blockage that is not prone to unwanted sediment flushing.

The bar screen had an inclination of 2:1 to favour the passage of driftwood over the barrier (Bezzola, Sigg, and Lange, 2004;

Lange and Bezzola, 2006; Piton and Recking, 2016b b). However, driftwood was not tested.

The combination of hydraulic and mechanical controls has shown to be promising in view of reducing risks due to individual uncertainties related to the unwanted sediment flushing and sediment size, respectively (Schwindt et al., 2016 b, 2017a). This combined control type was considered in the experiments (case *HyMec*) by the superposition of the bar screen to the hydraulic barrier with variable constriction height *a* and constant width *b*, according to the test case *Hy-o*.

**4.4 Experimental procedures**

Each barrier set-up was tested two times (*α* and *β* tests) with the same generic hydrograph that was established based on the following criteria:

- The duration of the falling limb $t_-$ (in s) is 1.7 times as long as the rising limb $t_+$ (in s), which is typical for floods of mountain rivers (Armanini and Larcher, 2001; D'Agostino and Lenzi, 1996; Kaitna et al., 2011; Piton and

Recking, 2016a; Rickenmann et al., 1998);



- The initial discharge of 5.5 l/s corresponds to the bankfull discharge of the guiding channel and the peak discharge of 12.5 l/s is imposed by the model limitations;

- The ratio between the sediment supply rate (bed load inflow $Q_{b,i}$) and the pump discharge $Q$ is 0.5 % (weight-specific), as determined in previous studies on the experimental set-up ( Schwindt et al., 2017b);

- The total supply volume $V_{\Sigma}$ (in m³) is higher than the plain storage volume (0.127 m³) of the deposition area (reservoir) considering a barrier height of 0.11 m.

The plain storage volume in the deposition area corresponded to the horizontal filling of the deposition area with a deposition slope $S_{dep} = 0$. The above-listed criteria led to a hydrograph with a rising limb duration of $t_+ = 1\ 129$ s ($\approx 19$ min) and a falling limb duration of $t_- = 1\ 920$ s ($\approx 32$ min). The water and solid discharge supply were adapted in steps of four minutes. The

10 resulting total volume of the sediment supply of the generic flood hydrograph was $V_{\Sigma} = 0.137$ m³. The hydrograph with sediment supply is shown in Figure 7 with the subsequently introduced dimensionless parameters.

At first sight, the ratio between peak and initial discharge of approximately 2.3 may seem low. However, the initial discharge represents the threshold value for triggering sediment deposition, i.e., a flood discharge that is potentially dangerous for downstream infrastructures. The peak discharge represents thus a flood that is in terms of magnitude by a factor of 2.3 higher

than the target discharge for triggering sediment retention.

Moreover, the possibility of sediment flushing was examined by trial of discharge variations, i.e., several sudden increases and decreases in the discharge were tested with the goal of attempting the remobilization of the deposit. The flushing attempts were only meaningful for the cases with hydraulic barriers, as the flushing of clogged mechanical barriers is not possible (Schwindt et al., 2017a). The duration of the flushing was based on the observation of the morphological activity in terms of sediment

displacements in the deposition area and the outflowing bed load.

## 4.5 Parameters and dimensional considerations

This study focuses on the deposition pattern and volume due to the standardized hydrograph, considering the occasional subsequent sediment flushing, and the corresponding transfer of bed load. These phenomena may be described by the following set $\Lambda$ of parameters:

$$\Lambda = \mathrm{f}\left(a, b, D_{84}, g, h, Q, Q_{b,i}, Q_{b,o}, S_o, t, t_+, t_-, V_{dep}, V_{\Sigma}, w, \nu, \rho_f, \rho_s, \rho_s{}'\right), \tag{1}$$

where a and b are the height and width of the hydraulic flow constrictions, respectively; $D_{84}$ is the representative grain size; g denotes the gravitational acceleration (9.81 m/s²); $h$ is the flow depth; $Q$ is the pump discharge; $Q_{b,i}$ and $Q_{b,o}$ denote the mass sediment supply and outflow rates, respectively; $S_o$ is the longitudinal slope of the guiding channel (5.5 %); $t$ is the experiment duration; $t_+$ and $t_-$ are the duration of the rising and falling limb of the hydrograph, respectively; $V_{dep}$ is the volume of sediment

deposits; $w$ is the channel bottom width; $\nu$ is the kinematic viscosity of water ($10^{-6}$ m²/s); $\rho_f$ and $\rho_s$ are the water density





(1 000 kg/m³) and the sediment grain density (2 680 kg/m³), respectively; and $\rho_s$' (1 550 kg/m³) is the density of sediment deposits, according to the supplier's data.

With respect to the analysis of bed load transport-related phenomena, the dimensional analysis is based on the independent variables of $D_{84}$, g and $\rho_f$ (Einstein, 1950; Yalin, 1977). The discharge $Q$ is considered relative to the bankfull discharge of the guiding channel ($Q_{bf} = 5.5$ l/s). In addition, the time $t$ is considered relative to the duration of the rising limb of the hydrograph; and the volume of sediment deposits $V_{dep}$ is considered relative to the cumulative volume of the hydrograph sediment supply ($V_\Sigma = 0.137$ m³). This leads to the following set of relevant dimensionless parameters:

- $a_* = a / D_{84}$, grain related opening height of vertical flow constrictions;

- $b_* = b / D_{84}$, grain related opening width of lateral flow constrictions;

- $Q_* = Q / Q_{bf}$, relative discharge;

- $s = \rho_s / \rho_f$, density ratio;

- $t_* = t / t_*$, relative duration;

- $V_* = V_{dep} / V_\Sigma \cdot 100$, percentaged relative deposit volume;

- $X_*, Y_*$ and $Z_*$ correspond to $x/D_{84}$, $y/D_{84}$ and $z/D_{84}$, respectively;

- $\Phi_i = Q_{b,i} / [w \cdot \rho_f \cdot ((s-1)\ g\ D_{84}{}^3)^{1/2}]$, intensity of bed load supply;

- $\Phi_o = Q_{b,o}\ [w \cdot \rho_f \cdot ((s-1)\ g\ D_{84}{}^3)^{1/2}]$, intensity of outflowing bed load.

Flow depth related parameters are not considered since the precise measurement of the flow depth was not possible by non-intrusive techniques in the shallow flow over the rapidly changing morphology of the sediment deposits during the hydrograph.

### 4.6 Summary of test runs

Table 1 lists the characteristic test parameters. The hydrograph was applied two times ($\alpha$ and $\beta$ tests) for every barrier configuration, except for the overflown hydraulic barrier (case *Hy-o*) as unwanted sediment flushing occurred in the first hydrograph test.

### 5. Results and Analysis

### 5.1 Evolution bed load transfer through the barrier

The outflowing sediment rates in terms of the bed load transport intensity $\Phi_o$ are shown in Figure 8 for the cases *Hy*, *Mec* and *Hy-Mec*, as a function of the relative hydrograph duration $t_*$ and for the two repetitive runs $\alpha$ and $\beta$. In addition, the shape of





the deposits at the peak of the hydrograph are shown in the top-view pictures. These pictures show the representative $\alpha$-tests, as no major differences between the pattern of the two repetitive tests ($\alpha$ and $\beta$) were observed.

In case *Hy-no* (Fig. 8 a), the outflowing bed load intensity $\Phi_o$ dropped in both tests ($\alpha$ and $\beta$) after a duration of approximately $t_* = 0.5$. This drop in $\Phi_o$ corresponds to the hydraulic clogging of the barrier. In parallel, the backwater of the infinitely high

barrier increased with increasing discharge ($t_* < 1$) and resulted in a regressive evolution of the sediment deposit in the upstream direction. The corresponding longitudinal evolution of the deposit is reflected in the top-view picture of the deposition area (Fig. 8 a) at the flood peak. Due to the influence of the deposit, the hydraulic jump could not migrate back in the downstream direction during the falling limb of the hydrograph ($t_* > 1$, Fig. 8 a). In consequence, the sediment flux through the barrier ceased with the flood peak ($\Phi_o = 0$) and the deposit laterally spread toward the banks of the deposition area at the end of the

hydrograph.

In case *Hy-o*, the relative constriction height $a_*$ was significantly smaller than previously (2.89 against 11.1 in case *Hy-no*). Therefore, nearby all of the supplied sediment was retained in the first half of the rising limb ($t_* < 0.5$, Fig. 8 a). Accordingly, the outflowing bed load intensity $\Phi_o$ decreased rapidly to zero, but $\Phi_o$ restarted to increase with the second increase of the discharge. The raise in the discharge (cf. Fig. 7) at $t_* = 0.37$ corresponds to an increase from $Q = 7$ l/s to $Q = 8.5$ l/s, i.e., the

15 desired threshold value for initiating the barrier overflow. As it can be observed in the top-view picture (lower top-view picture on Fig. 8 a), the sediment flushing started already before the flood peak ($t_* < 1$). After the flood peak ($t_* \geq 1$), the flushing of nearby all the previously deposited sediment occurred. The observed maximum of $\Phi_o = 0.32$ during the flushing corresponds to approximately 1.4 times the maximum supply rate of $\Phi_i = 0.23$ at the flood peak. A repetitive run of this configuration was discarded due to the unwanted sediment flushing observed before the flood peak. In practice, every barrier can be overflown

when the discharge is high enough. However, the comparison of the cases *Hy-no* and *Hy-o* shows that barriers for only hydraulic control need to be sufficiently high to avoid such unwanted sediment flushing. Even though reducing the dimensions of the opening in the barrier could increase the safety against self-flushing, smaller constriction heights or widths were not tested to avoid sediment retention before the bankfull discharge of the guiding channel (5.5 l/s) was reached.

In case *Mec* (Fig. 8 b), the temporal evolution of the outflowing bed load intensity $\Phi_o$ was similar to the supply intensity $\Phi_i$

(cf. Fig. 7) until the flood peak occurred ($t_* = 1$). Hence, only marginal sediment deposits close to the barrier can be observed in the top-view picture of the deposition area at the flood peak. At a relative flood duration of approximately $t_* < 1.25$, the bar screen was mechanically clogged, and consequently, the outflowing bed load intensity $\Phi_o$ decreased in both tests ($\alpha$ and $\beta$) to zero by stages. An elongated deposit in the deposition area was observed at the end of the hydrograph according to descriptions from Campisano et al., 2014 and Piton and Recking, 2016a.

In case *HyMec* (Fig. 8 c), the outflowing bed load intensity $\Phi_o$ decreased rapidly to zero for the smaller opening heights $a_1$ and $a_2$. With the largest opening height $a_3$, $\Phi_o$ was similar to the supply intensity $\Phi_i$ at the beginning. Only with the second increment of the discharge and sediment supply at $t_* = 0.37$, the barrier clogged. After the barrier clogging, an elongated deposit developed layer-wise until it reached the barrier height at $t_* \approx 0.6$ for the three considered constriction heights. In consequence, the supplied sediment was transported over the barrier, which is reflected in the evolution of the outflowing bed load intensity





$\Phi_o$ that corresponds to the supply intensity $\Phi_i$ (cf. Fig. 7). However, $\Phi_o$ is slightly smaller than $\Phi_i$, as the deposit enlarged after $t_\star = 0.6$. This enlarged deposit shape can be observed in the corresponding top-view picture of the deposition area (Fig. 8 c). The repetitive tests ($\alpha$ and $\beta$) resulted in similar outflow rates for the three opening heights.

A major difference in $\Phi_o$ can be observed in the test *HyMec.a₂ β*, where a constant discharge of 5.5 l/s with sediment supply
was applied prior to the hydrograph, for a duration corresponding to $t_\star$, i.e., $t_\star \approx 1$. This combination of low discharge and sediment supply led to the decelerated clogging of the combined barrier. The consequence was an early evolution of the backwater in upstream direction, beyond the upper limit of the observation reach resulting in an almost total retention of the sediment supply.

## 5.2 Volume of sediment deposits in the deposition area

The volumes of the sediment deposits were measured by three redundant tools, namely the laser, the motion sensing camera (Kinect) and the total weight of the deposited sediment measured with the industrial scale. This redundant evaluation was necessary because the scale gives only information about the sediment weight and the bathymetric data from the motion sensing camera and laser can contain individual measurement errors (Lachat et al., 2015). The motion sensing camera provides a high-resolution bathymetric image of the deposit, but the image required a correction due to distortion and the surface texture. The
laser measurements are precise but the point density is low, which leads to averaging errors in the surface interpolation. For the determination of the deposit volume with both approaches, the bathymetric surface data of the empty deposition area were subtracted from the surface data of the sediment deposits. An example application of the bathymetric recording of the deposit with the motion sensing camera after the test *HyMec.a₁* α is shown in Figure 9. The bathymetric deposit volume $V_{dep}$ (Bathymetric) according to both the camera and the laser was then determined using CAD software.
After every hydrograph test, the deposited sediments were flushed (without any barrier) in the filter basket which was weighed with the industrial scale. This weight was divided by the deposit density $\rho_s'$ of 1 550 kg/m³ to obtain the according deposit volume $V_{dep}$ (Scale). The comparison of $V_{dep}$ (Scale) and $V_{dep}$ (Bathymetric) was used to evaluate the percentaged error $\varepsilon_v$ of the bathymetric tools (except for the case *Hy–no*, where sediment flushing was examined after the hydrograph).

$$\varepsilon_v = \left[V_{dep}(Bathymetric) - V_{dep}(Scale)\right]/V_{dep}(Scale) \cdot 100, \qquad\qquad (2)$$

The error $\varepsilon_v$ is shown in Figure 10 for the cases *Mec* and *HyMec*, where the bar screen was applied. The graphs show that both bathymetric techniques tend to underestimate the deposit, but this effect is significantly less pronounced for the camera data (in average, $\varepsilon_v = 2.7\,\%$) than for the laser data (in average, $\varepsilon_v = 14.8\,\%$).

The complex application of the centimetre-wise laser measurements was restricted to 16 profiles (approximately 650 points), and therefore, it is less precise than the camera data (mm-wise, $1.92 \cdot 10^6$ points). Hence, the motion sensing camera is
subsequently used for the analysis of the deposit pattern.

The relative deposit volumes $V_\star$, i.e., the ratio of the deposit volumes $V_{dep}$ and the supply volumes $V_\Sigma$, are shown in Figure 11 based on the scale measurements as a function of the test cases. As expected from the results regarding the sediment outflow



rates (cf. Fig. 8), the total deposit volume is very small in the case *Hy-o*, while it is high in the test *HyMec.a₂β*. The case *Hy-no* was not evaluated because sediment flushing with additional sediment supply was tested after the hydrograph. However, the graphs of the bed load intensity $\Phi_o$ (Fig. 8 a) indicate that $V_*$ is close to 100 % in the case *Hy-no*. The relative deposit volume $V_*$ varied in the cases *Mec* and *HyMec* between approximately 40 and 55 %, invariant of the presence of the bar screen.

5 In these cases (*Mec* and *HyMec*), $V_*$ refers to the backwater-driven storage space upstream of the clogged barrier without the occupation of the entire width of the deposition area. This indicates that the barrier height is essential for the amounts of retained sediment, independent from the control type (mechanical and/or hydraulic). However, the moment of the barrier clogging, as a function of $t_*$, is important for the attenuation of sediment peak flows, as the comparison between Fig. 8 b) and Fig. 8 c) shows.

## 10 5.3 Deposition patterns

The final shapes of the sediment deposits were recorded at the end of every hydrograph test. According to the evolution of the sediment outflow (cf. Fig. 8), the deposition patterns of the repetitive α and β-tests were almost similar. Therefore, the deposition patterns obtained by the motion sensing camera are compared in Figure 12 with top-view pictures, only for the α-tests. Moreover, only one representative plot (test *HyMec.a₃ α*) of the relative deposit height $Z_*$ is shown for the three

constriction heights applied in the case *HyMec*, as the constriction height variation had no measurable effect on the sediment deposit.

Similar to the sediment outflow rates (cf. Fig. 8) and relative deposit volumes (cf. Fig. 11), it can be observed that the deposit was wide and deep in the case *Hy-no*. The deposition patterns of the cases *Mec* (mechanical barrier only) and *HyMec* (combined barrier) differed only marginally.

According to the relative retention volumes $V_*$ (cf. Fig. 11), the volume and deposition pattern differences between the tests *HyMec.a₁ α* and *HyMec.a₃ β* are small. Both tests corresponded to the minimum and maximum tested constriction heights $a_1$ and $a_3$, respectively. In addition, the deposit height was slightly lower in the tests *HyMec.a₁ β*, *HyMec.a₂ α* and *HyMec.a₃ α*. These observations indicate that there is no evident effect of the (relative) constriction height on the deposition pattern within the tested range of $a_*(min) = 2.89$ and $a_*(max) = 3.44$. Moreover, this observation is in agreement with the sediment outflow

rates (cf. Fig. 8 c), where the time variation curves of $\Phi_o$ are very close to each other.

The deposition pattern after the *Hy-o*–test was not recorded, as there were only small sediment remainders on the overbanks, as shown in Figure 13.

## 5.4 Sediment flushing

Figure 14 shows the evolution of the outflowing bed load intensity $\Phi_o$ for the flushing in the case *Hy-no* (non-overflown flow

constriction) after the hydrograph tests α and β, as a function of the multiple duration $t_*$ of the hydrograph rising limb. Although, similar tests were run for the case *HyMec.a₂* (combined barrier), these results are not shown here because it was impossible to remobilize sediments from the deposit ($\Phi_o$ is a horizontal zero-line).





The technically maximum possible sediment volume (model limit: 727 kg) was supplied at the beginning, followed by a phase of clear water flow for both flushing attempts ($\alpha$ and $\beta$). The flushing of test *Hy-no α* showed some sheet-wise grain mobilizations from the deposit between $t_* = 2.5$ and $t_* = 3.5$ when the discharge was decreased (Fig. 14, *Hy-no α*). Only minor morphological activity was observed after the discharge decrease. Also a sudden, arbitrary increase in the discharge with subsequent decrease toward the end of the experiment did not remobilize the grains. The flushing of test *Hy-no α* was stopped after a duration of more than 12 times the rising limb of the hydrograph, as no further morphological activity was observed.

The flushing of the test *Hy-no β* continued for 26 times the duration of the rising limb of the hydrograph, with several trials of discharge variations. Similar to the $\alpha$-test, the maximum possible sediment volume was supplied at the beginning. After every step of discharge decrease, the sheet-wise flushing of sediment from the tip of the deposit was observed. The maximum of these flushings reached an outflow intensity $\Phi_o$ corresponding to the supply peak of the hydrograph (Fig. 14, *Hy-no β* and Fig. 7). These flushings were mainly observed when the discharge conditions in the flow constriction changed from pressurized to free surface flow.

Toward the end of the β-test, from $t_* \approx 22$ to $t_* \approx 23$, an attempt was made to induce the flushing of the guiding channel. This was achieved by the experiential, successive removal of the upper layer of the deposit along the axis of guiding channel. The experientially created depression had a depth of approximately $2 \cdot D_{84}$ and a width of approximately 0.1 m, corresponding to the bottom width of the guiding channel. This experiential depression was created stepwise, beginning at the tip of the deposit (downstream end), then continuing the excavation in the upstream direction. However, only marginal morphological activity was observed, unless the tail of the deposit (upstream end), i.e., the hydraulic jump, was directly connected with the opening through the depression. Small meanderings were observed at the beginning of the flushing through the depression (Figfure 15 a-c). In the following, the depression incised from the upstream toward the downstream direction (Fig. 15 d-e), until the guiding channel was completely cleared (Fig. 15 f). The relative discharge during the flushing of the guiding channel was $Q_* = 1.2$, i.e., $Q = 1.2 \cdot Q_{bf}$. A comparison of the maximum sediment outflow intensity $\Phi_o$ with the Smart and Jaeggi (1983) formula applied to the geometry of the guiding channel showed good agreement as already observed in previous studies (Schwindt et al., 2017c).

## 6. Discussion

### 6.1 Sediment deposition

The elongated deposits at the end of the hydrograph tests were typical for the overflown barrier (cf. Fig. 12 b and c), where the deposition control functioned as desired without unwanted flushing (*Mec* and *HyMec*). The high, non-overflown barrier (*Hy-no*) caused a wider and longer spread of the deposit (cf. Fig. 12 a), which is in agreement with the observations from Zollinger (1983). The storage volume upstream of overflown barriers may increase when the deposition slope $S_{dep}$ is additionally considered. $S_{dep}$ can be estimated as a function of the channel slope $S_o$ and it is typically in the range of $1/2 \cdot S_o$ for small floods and $2/3 \cdot S_o$ for large floods with high sediment concentration (D'Agostino, 2013; Osti and Egashira, 2013; Piton





and Recking, 2016a a). The deposition slopes observed in the present study can be obtained by the relative deposit height $Z_*$ at the longitudinal section at the axis of the guiding channel ($Y_* = 0$). Linear regression curves have been established in Figure 16 to estimate $Z_*$ as a function of $X_*$ in the empirically determined aggradation zone upstream of the barriers. Thus, the slope of the regression curves corresponds to the deposition slope $S_{dep}$ in the considered aggradation zones. This evaluation

results in $S_{dep}(Hy\text{-}no) = 6.5\,\%$, $S_{dep}(Mec) = 12\,\%$ and $S_{dep}(HyMec) = 9.5\,\%$. Compared with the bottom slope $S_o$ of the guiding channel, these values correspond to $S_{dep}(Hy\text{-}no) = [1–2]\cdot S_o$, which is significantly higher than the values corresponding to the above mentioned literature.

The deposition slope can also be approached using the equilibrium slope, assuming that the sediment supply and erosion are balanced on a reach scale. Zollinger (1983) proposed to apply the Smart and Jaeggi (1983) formula with respect to zero-

10 transport conditions ($\Phi = 0$). This approach was not possible for the experiments, as the clear water depth was highly variable and not measurable due to the shallow flow over the changing sediment deposits. As an alternative, a relationship for the equilibrium slope was applied, as proposed by (Johnson, 2016):

$$S_{(Johnson,2016)} = \frac{c \cdot w}{Q} \cdot D_{84}^{3/2} \cdot (s-1) \cdot \left[\left(\frac{\Phi_o}{3.97}\right)^{2/3} + \tau_{*cr}\right]^{3/2}, \tag{3}$$

Eq. 3 was evaluated by using the peak discharge of the hydrograph and the bed load transport intensity over the barrier

($HyMec$). The width $w$ was substituted by the barrier spill width of 0.234 m and a value of 0.05 was considered for the dimensionless bed shear stress $\tau_{,cr}$. This results in equilibrium slopes between 12 and 15 % for the $HyMec$-tests. Applying Eq. 3 at the instant when the sediment transport across the barrier ceased, results in very small values of $S_{dep} < 1\,\%$. Thus, Eq. 3 is not appropriate for estimating the deposition slope. In practice, it is safer to assume small values of the deposition slope for estimating the maximum storage upstream of the barrier. Such a safe estimate can be made by the relationship $S_{dep} = 1/2 \cdot S_o$.

The deposit shape, independent of the barrier height and type, is in practice often confined by the terrain morphology. Thus, the deposition area of such confined sediment traps corresponds to the riverbed and its overbanks. Such elongated, natural deposition areas are more exposed to sediment flushing because of the higher concentration of the stream power over the width of the deposition area (Leys, 1976; Zollinger, 1983).

### 6.2 Sediment flushing

The flushing of the non-overflown barrier ($Hy\text{-}no$) was not possible without artificial intervention. However, the overflown hydraulic barrier ($Hy\text{-}o$) is prone to unwanted flushing, as it was observed during the hydrograph. The safety against unwanted flushing through such overflown permeable barriers may also be increased by reducing the dimensions of the opening, but smaller constriction dimensions are not favourable regarding the eco–morphological river continuity. Thus, the application of permeable barriers with very limited height for solely the hydraulic control of bed load retention is not recommendable for the

practice.



The height of the overflown permeable barrier in the case *Hy-o* corresponded to the theoretic cross-section averaged energy head (clear water flow) in the guiding channel with respect to the target discharge for the initiation of overspill of the barrier. Naturally, these observations show that the maximum possible backwater depth caused by such barriers is a decisive factor for the reduction of the energy slope upstream of the barrier. For this purpose, former studies considered only the dimensions of

5 the opening in the barrier (Schwindt et al., 2017c) but not the barrier height. The present study differentiates only between infinite and limited barrier heights, but indicates that future works need to consider systematically the influence of the barrier height on sediment flushing through hydraulic control openings.

The sediment flushing through the mechanically clogged bar screen was impossible, as shown by the attempts after the *HyMec.a₂* hydrograph-tests. The flushing attempts through the non-overflown hydraulic barrier (*Hy-no*) have shown that the

10 tip of the deposit repetitively collapses, when the flow conditions in the opening of the barrier pass from pressurized to free surface flow. Such observations were already made in earlier studies (Zeller, 1973).

According to previous studies, the flushing processes of sediment traps is a succession of the discharge-driven reshaping of a network of sub-channels in the deposition area. The continuous reshaping lead to a gradual incision of the deposit along the longitudinal axis of the initial riverbed (Armanini and Larcher, 2001; Busnelli et al., 2001; Piton and Recking, 2016a; Zollinger,

1983). This observation was not made in the present study, as apparent grain imbrication caused the armouring of the surface layer of the deposit. Only a trial of artificial breaking of the armouring layer along the longitudinal axis of the guiding channel enabled sediment flushing (cf. Fig. 15). The subsequent morphological activity caused further incision of the initiated channel, with only little meandering. Once the guiding channel was cleared, no further lateral or vertical erosion was possible. Thus, the guiding channel directs not only sediment-laden flows through the sediment trap up to small flood discharges for which no

sediment retention is required, but it also enables the controlled, desired flushing of previously retained sediments through a hydraulic control barrier. The triggering of such desired sediment flushing requires the prior removal of mechanical logjams. The remaining deposits have to be excavated and may be replenished downstream at suitable locations for improving sediment transport dynamics (Battisacco et al., 2016).

### 6.3 Eco-morphological aspects

The guiding channel enables the undisturbed conveyance of sediment-laden (flood) discharges until its bankfull discharge is reached. Therefore, the opening in the hydraulic barrier should not affect the flow before the bankfull discharge of the guiding channel is reached. Previously established formulae for estimating the discharge capacity of the opening in the hydraulic barrier can be used to determine the extent of backwater due to the barrier (Armanini et al., 2006; Armanini and Larcher, 2001; D'Agostino, 2013; Piton and Recking, 2016a; Schwindt et al., 2017b). These formulae consider upstream flow conditions, i.e.,

the flow conditions in the guiding channel, and can be used to design the opening in a way that it does not cause backwater until the bankfull discharge of the guiding channel is reached. Thus, the opening width should at least correspond to the bottom width of the guiding channel.




The guiding channel should be designed based on the dominant, morphologically effective discharge in view of the dynamic evolution of downstream reaches. Moreover, the guiding channel should provide appropriate hydraulic conditions for fish migration, in terms of the required flow depth and maximum velocity (Baigún et al., 2012; DWA, 2014; Gisen et al., 2017; Tamagni, 2013). This can be achieved through a nature-oriented trapezoidal cross-section geometry, with a rough channel

bottom constituted by large distributed boulders and a sufficient channel width.

For a natural eco–morphological diversity of downstream reaches, also driftwood is important (Gilvear et al., 2013; Senter and Pasternack, 2011). However, the retention of driftwood is sometimes necessary when trunks or rootstocks cannot pass downstream bottlenecks at urbanized river reaches (Lassettre and Kondolf, 2012; Mazzorana et al., 2012). Appropriate measures for driftwood retention were proposed and discussed by Comiti et al., 2012, Lange and Bezzola, 2006; Schmocker

and Weitbrecht, 2013.

### 6.4 Application and limits

Piton and Recking (2016a) present a 13-step approach for the design of sediment traps and check dams: Steps 1-3 describe the identification of relevant torrential hazards, the structure location and retention objectives. Steps 4-11 represent an iterative design of the shape, size and bottom slope of the deposition area combined with an open check dam. Steps 12 and 13 address

the design of spillways and scour protection measures. In this framework, this study relates to the iterative design of the deposition area and the open check dam, where the implementation of a guiding channel is additionally recommended (i.e., after step 5 in Piton and Recking, 2016a). The verification of the retention objectives in terms of mechanical clogging and the hydraulic functionality in terms of the discharge capacity and local head losses (steps 8 and 9 in Piton and Recking, 2016a) in the present study is similar to the one presented in Piton and Recking (2016a). However, the functionality in terms of

mechanical clogging and hydraulically induced sediment retention is triggered in this study by two different measures in the shape of a bar screen and an open check dam. Therefore, both elements require differentiated verification of their functionality. The dominant discharge can be very high in strongly armoured mountain rivers or channels confined by bedrock outcrops (Hassan et al., 2014). In such rivers, it may be preferable to forgo the permeability of sediment traps, as the transport of sediment is related to exceptional floods. In these cases, the installation of barriers combining mechanical and hydraulic

controls, as discussed here, is also advantageous to ensure the fail-safe sediment retention. Then the design of the barrier should refer to the sediment characteristics of the catchment area and the flood discharge which potentially endangers urban downstream reaches.

The small ratio of 2.3 between the peak discharge and the initial discharge of the flood hydrograph seems low. It can be argued that an annual flood is not yet considerable for triggering sediment retention. Therefore, the bankfull discharge of the guiding

channel should be higher than an annual flood. Thus, regarding the experimental study, an annual flood should be smaller than the initial discharge of the flood hydrograph. The 132 dataset used for this study (Schwindt, 2017) show an average a ratio between a 100-year and an annual flood discharge of approximately 2.8. In this context, the applied ratio of 2.3 between initial and peak discharge in the experiments can be considered plausible.



## 7. Conclusions

The concept of typical sediment traps, consisting of a widened deposition area with downstream deposition control barrier, is enhanced by a guiding channel and tested with different partially open barrier types.

The guiding channel ensures that sediments are transported through the deposition area, without any deposition, up to its bankfull discharge. Moreover, the guiding channel serves for the flow control in the deposition area, which is important to ensure the desired functioning of the permeable barrier.

The open barrier needs to be designed for sediment retention once the bankfull discharge of the guiding channel is exceeded. The sediment retention due to the barrier is differentiated here between hydraulic and mechanical controls, as well as the combination of both.

This experimental study of the guiding channel combined with the barrier for hydraulic and/or mechanical controls, based on a generic hydrograph with occasional, subsequent flushing shows that:

- The guiding channel fulfils its purpose of promoting the river continuity until its bankfull discharge is exceeded;

- Overflown barriers with hydraulic control only are susceptible to unwanted sediment flushing during floods;

- The fail-safe obstruction of open barriers can be achieved by combining the hydraulic and mechanical controls to compensate individual risks related to unwanted sediment flushing and the grain size;

- Partial, desired sediment flushing through hydraulic control barriers after a flood can be artificially enabled.

### Acknowledgment

This work is funded by the Swiss Federal Office for the Environment (SFOEN) under the Sediment and Habitat Dynamics research project. Giorgio Rosatti and Guillaume Piton contributed with the constructive exchange of ideas. Further thanks goes to Fritz Zollinger, Daniela Nussle and Gian Reto Bezzola for their kind acceptance to use their illustrations as model for figures.

## Nomenclature

| | |
|---|---|
| $A$ | Flow cross section (m²) |
| $a_*$ | Relative constriction height (–) |
| $b$ | Constriction width (m) |
| $b_*$ | Relative constriction width (–) |





| | |
|---|---|
| $C$ | Chézy flow resistance coefficient ($m^{1/2}$ $s^{-1}$) |
| $D_{xy}$ | Grain diameter of which xy % of the mixture are finer (m) |
| $g$ | Gravity acceleration (m $s^{-2}$) |
| $m$ | Channel bank slope (–) |
| $p1/p2$ | Coefficients of linear regression curve (–) |
| $Q$ | Water discharge (m³ $s^{-1}$) |
| $Q_*$ | Discharge relative to bankfull channel capacity (–) |
| $Q_{b,i}$ | Bed load supply rate (kg $s^{-1}$) |
| $Q_{b,o}$ | Bed load outflow rate (kg $s^{-1}$) |
| $S_{dep}$ | Deposition slope (–) |
| $s$ | Ratio of grain and water density (–) |
| $t_+$ | Duration of rising hydrograph limb (s) |
| $t_-$ | Duration of falling hydrograph limb (s) |
| $t_*$ | Duration, relative to the rising hydrograph limb (–) |
| $V_*$ | Percentaged deposit volume, relative to hydrograph supply (%) |
| $V_{dep}$ | Volume of sediment deposits (m³) |
| $V_\Sigma$ | Sediment supply volume during hydrograph (m³) |
| $w$ | Channel bottom width (m) |
| $x$ | Channel axis, pointing in the upstream direction (m) |
| $X_*$ | Dimensionless channel axis (–) |
| $y$ | Lateral axis, pointing toward the right bank (m) |
| $Y_*$ | Dimensionless lateral axis (–) |
| $z$ | Vertical axis, pointing against gravity acceleration vector (m) |
| $Z_*$ | Dimensionless vertical axis (–) |
| $\varepsilon_v$ | Percentaged error of the volume measurements (%) |
| $\Phi_i$ | Bed load supply intensity (–) |
| $\Phi_o$ | Outflowing bed load transport intensity (–) |
| $v$ | Kinematic viscosity (m² $s^{-1}$) |
| $\rho_f$ | Water density (kg $m^{-3}$) |
| $\rho_s$ | Grain density (kg $m^{-3}$) |
| $\rho'_s$ | Deposit density (kg $m^{-3}$) |
| $\tau_{*,cr}$ | Shields-parameter (–) |




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

**Figure 1: Concept of a permeable sediment trap consisting of (1) an open barrier (open sediment check dam) with overflow crest for flood release, followed by (2) downstream abutments with counter dam (sill); (3) a reservoir or deposition area, limited by (4) lateral dykes; (5) a maintenance access; and (6) an inlet structure with scour protection (adapted from Piton and Recking, 2016a; Zollinger, 1983). For permeable sediment traps, the novel element of (A) a guiding channel is introduced with (B) a barrier consisting of a bar screen for mechanical control and a barrier with an opening for the hydraulic control of bed load retention (Schwindt et al., 2017a).**

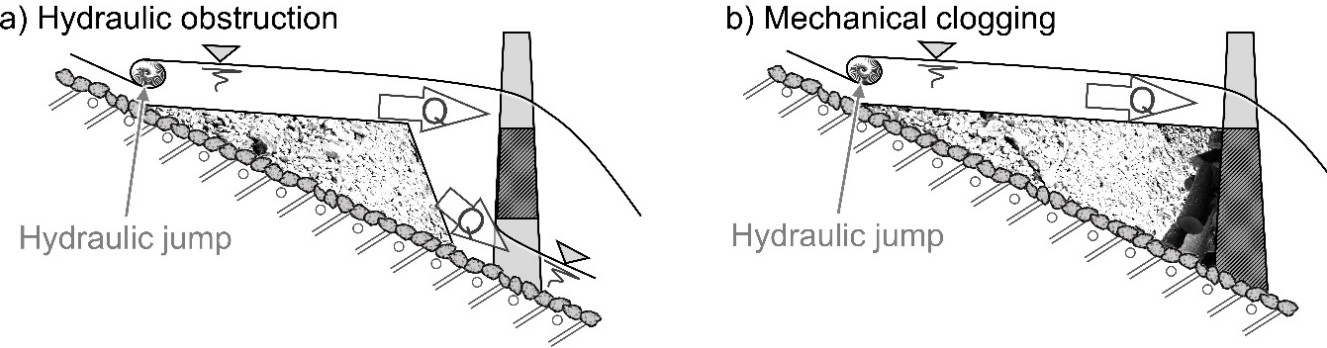

**Figure 2: Control mechanisms of sediment retention by permeable torrential barriers: a) hydraulic deposition, where the opening creates backwater due to the exceedance of its discharge capacity, and b) mechanical deposition caused by large objects (adapted from Lange and Bezzola, 2006; Piton and Recking, 2016a).**

Natural Hazards
and Earth System
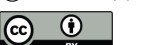


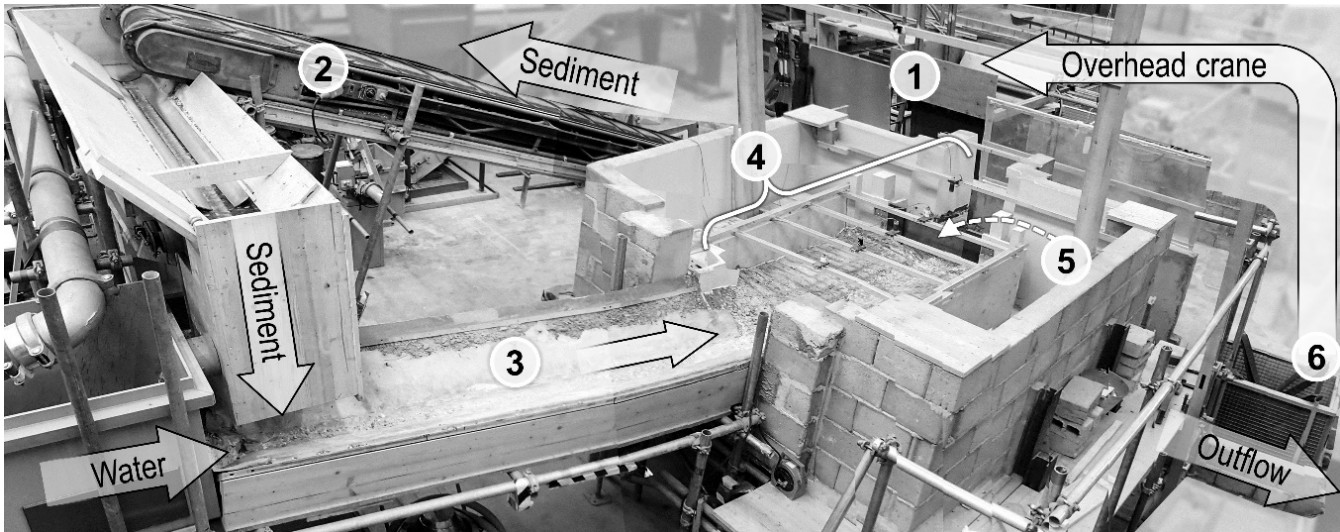

**Figure 3: The experimental set-up with sediment supply system that consisted of a sediment container (1) and conveyor belts (2); with indication of the water supply by the laboratory pump system, and the adaptation reach (3) that lead the sediment-water mixture to the observation reach (4). The barriers were placed at the downstream end of the observation reach (5). The outflowing sediment and water were separated by a filter basket (6) at the downstream model end.**

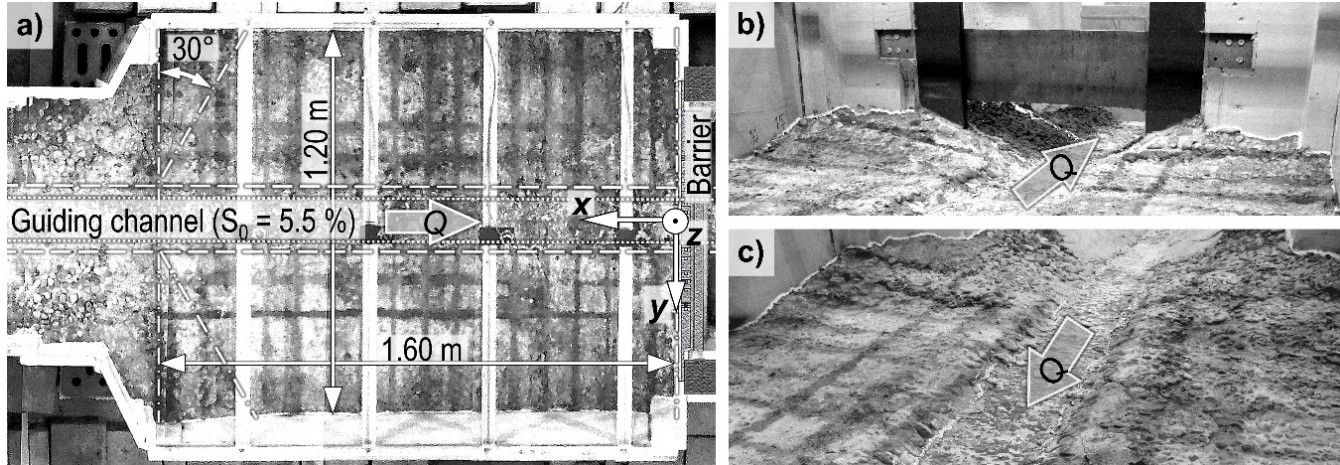

**Figure 4: Details of the observation reach consisting of the deposition area (reservoir) with guiding channel. The marked grid lines on the bottom were used for qualitative purposes and had an interspace of approximately 0.1 m: a) top-view with indication of the reservoir length (1.60 m), width (1.20 m), opening angle (30°) and longitudinal slope (5.5 %), as well as the model coordinate system ($x, y, z$ axis), used for the evaluation of sediment deposits; b) location of barriers, view in the downstream direction; and c) deposition area (reservoir), view in the upstream direction.**



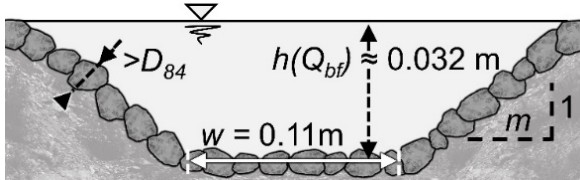

**Figure 5: The cross-section of the trapezoidal guiding channel, lined with fixed grains larger than the D84 of the sediment supply mixture and designed for bank overtopping for discharges higher than 5.5 l/s.**

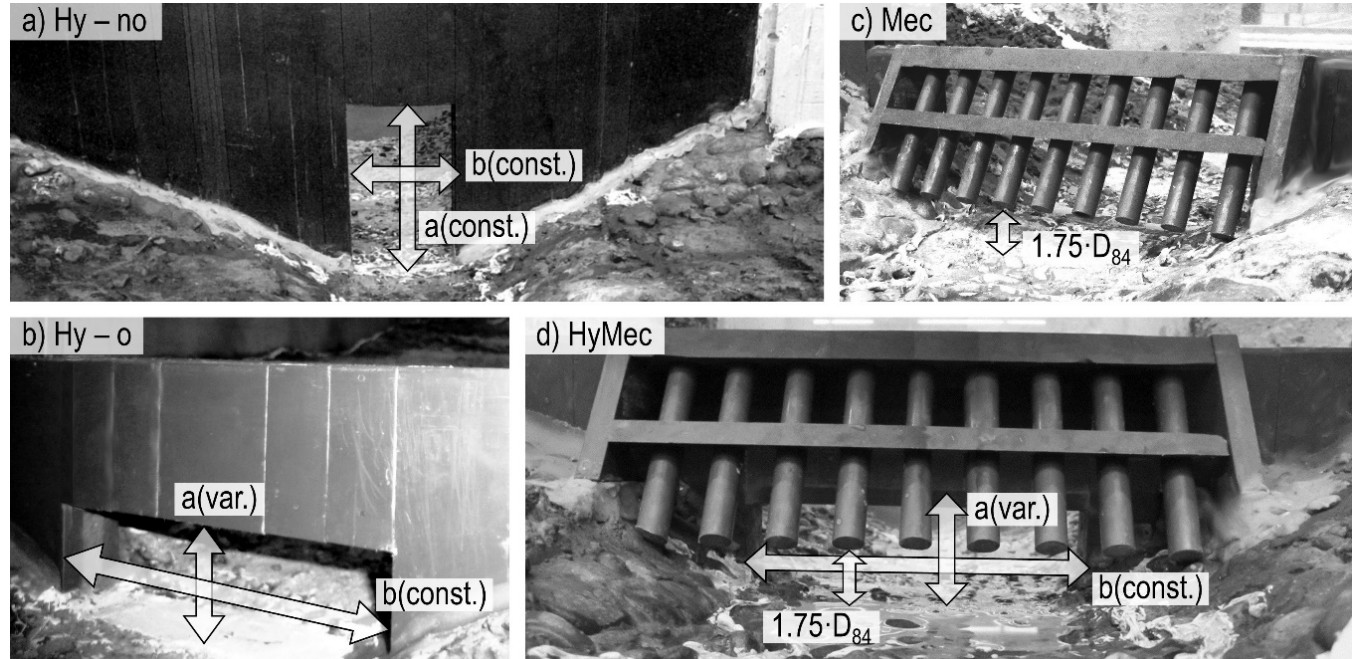

**Figure 6: Tested barrier types: Hydraulic deposition control only with constriction height a and width b; a) case *Hy-no* without the possibility of structure overflow and b) case *Hy-o*, with limited barrier height (0.11 m); mechanical deposition control by c) a bar screen (case *Mec*) with a height of 0.11 m; and d) the combination of hydraulic and mechanical deposition control (case *HyMec*), with the bar screen superposed to the flow constriction with variable constriction height a and constant width *b*.**





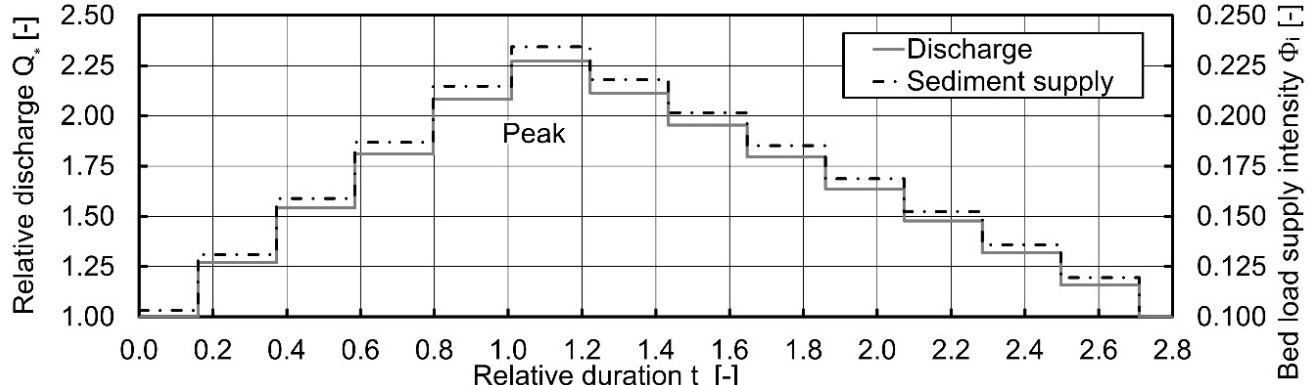

**Figure 7: The generic hydrograph used for the experiments, based on the dimensionless expressions of relative discharge $Q_* = Q / Q_{bf}$, bed load supply intensity $\Phi_i$ and the relative time $t_* = t / t_*$.**



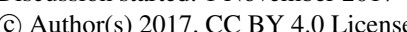

**Figure 8: The outflowing bed load transport intensity $\Phi_o$ as a of function the relative time $t_*$ and for the two repetitive tests $\alpha$ and $\beta$.** a) for hydraulic control without barrier overflow (*Hy-no*) and with barrier overflow (*Hy-o*); b) for mechanical control by the bar screen (*Mec*); and c) for combined deposition controls (*HyMec*), i.e., the combination of hydraulic barrier with varying opening heights $a_{1,2,3}$ and upstream superposed bar screen. The top-view pictures at the right show the sediment deposits at the flood peak of the $\alpha$-tests.


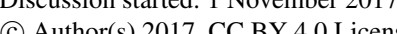



**Figure 9: Example of the recording of the deposition area bathymetry with the motion sensing camera: (a) a gray-scale picture of the empty deposition area (top-view) and (b) a gray-scale picture of the deposition area with sediment (top-view). A picture from a standard camera of the deposit at the end of the _HyMec.a1_ α-test is shown in (c), with its numerical representation derived from the motion sensing camera (d).**





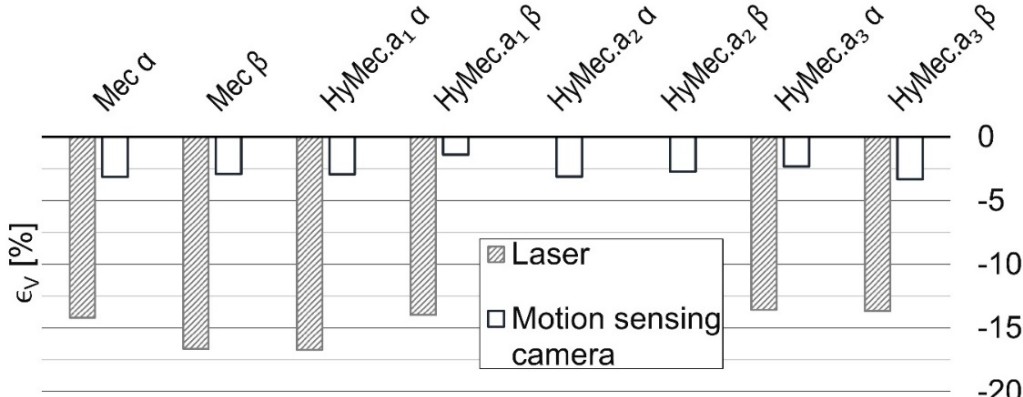

**Figure 10: Percentage error $\varepsilon_v$ of the sediment volume derived from weight measurements (assuming $\rho_s' = 1\,550$ kg/m³) and the deposit volume measurements based on the bathymetric scans using the laser and the motion sensing camera; the bathymetric records were made after the repetitive $\alpha$ and $\beta$ tests with the bar screen only (*Mec*) and the combination of the bar screen with the open hydraulic barrier *HyMec*, with varying opening heights $a_{1,2,3}$.**

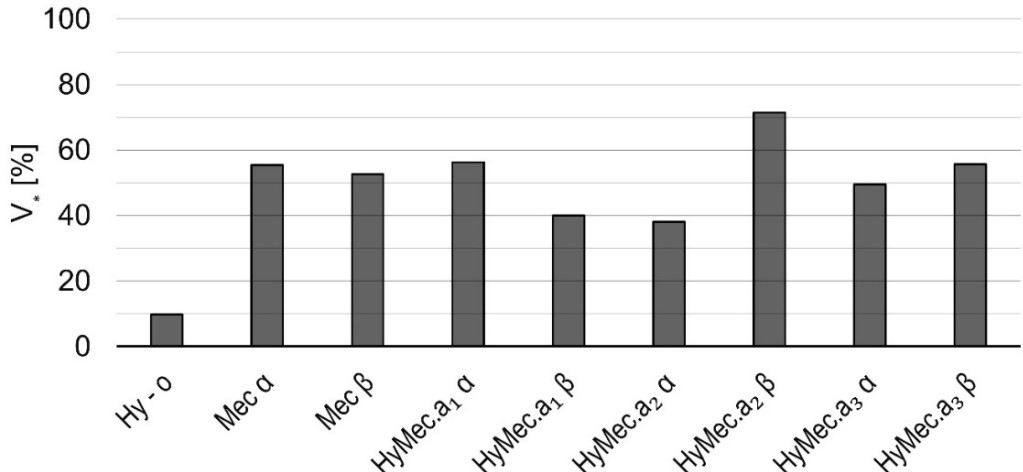

**Figure 11: The ratio $V_*$ (in %) of the deposit volume $V_{dep}$ and the supply volume $V_\Sigma$ after the repetitive hydrograph tests $\alpha$ and $\beta$ for the cases of the non-overflown flow constriction (*Hy-no*), overflown bar screen (*Mec*) and the combination of overflown bar screen superposed to the flow constriction (*HyMec*), with varying opening heights $a_{1,2,3}$.**



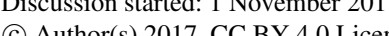

**Figure 12: Deposition patterns at the end of the hydrograph tests; left column: top-view pictures; right column: bathymetric records, a) in case *Hy-no* (*a*-test), with non-overflown hydraulic barrier; b) in case Mec (*a*-test), with bar screen for mechanical control only; and c) case HyMec (test *a₃a*), with combined hydraulic barrier and upstream superposed bar screen.**





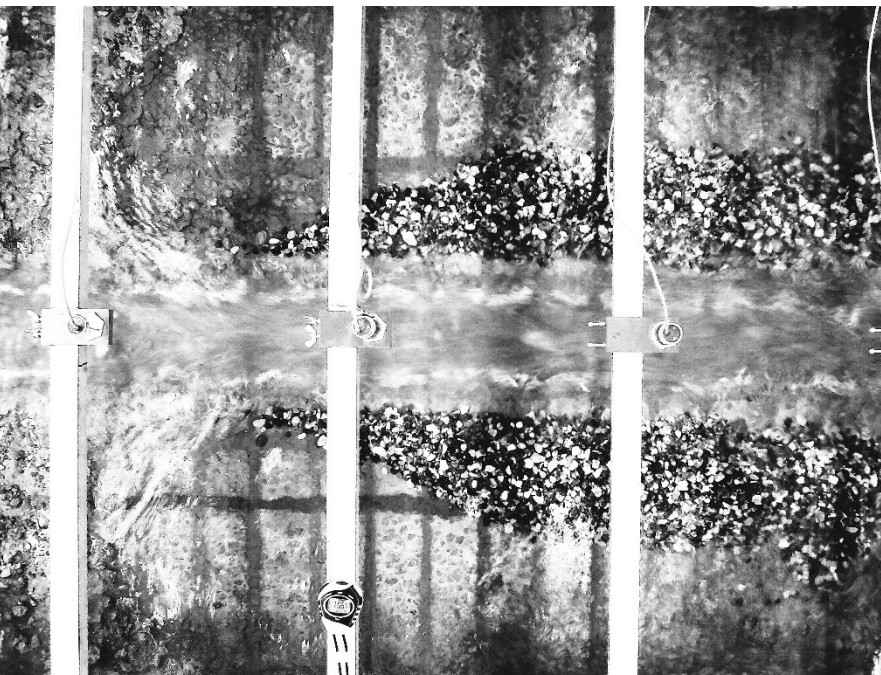

**Figure 13: Remaining sediment deposits at the end of the hydrograph test of case *Hy-o*.**





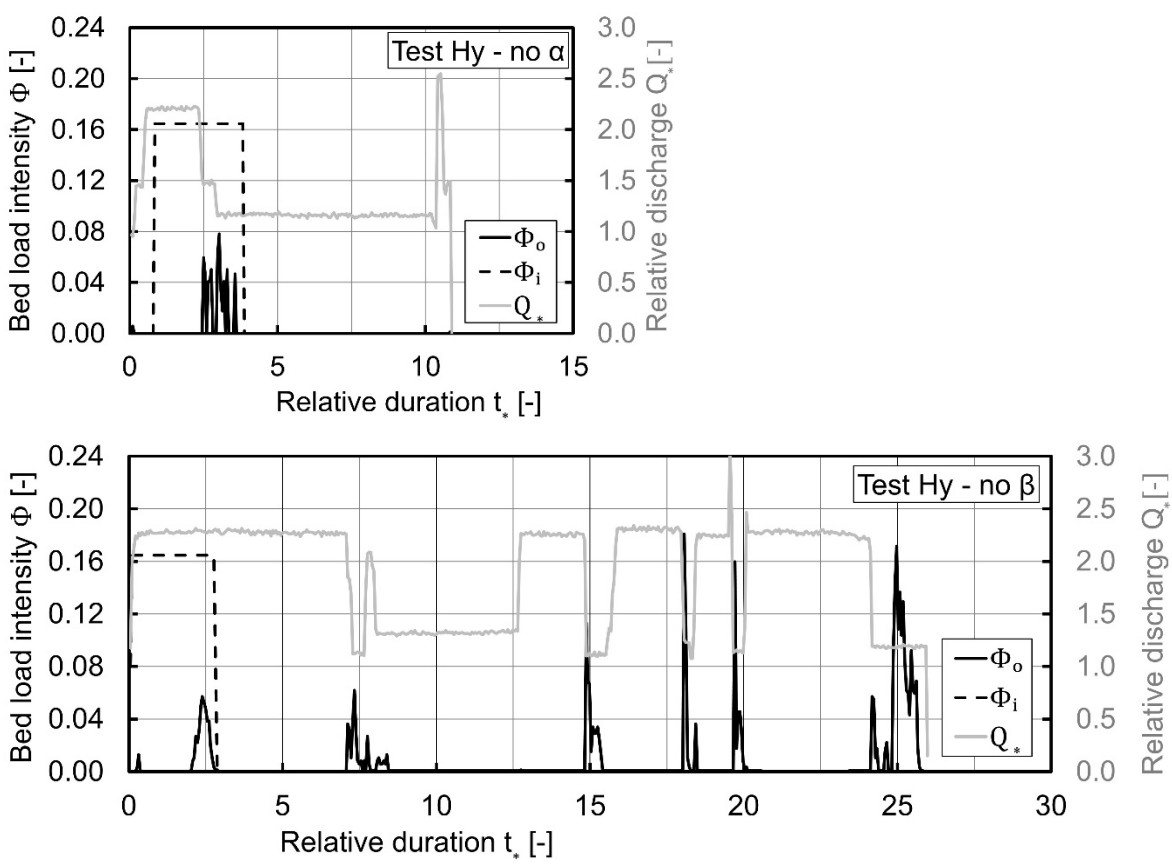

**Figure 14: Evolution of the outflowing bed load intensity $\Phi_o$ for the sediment flushing attempts after the $\alpha$ and $\beta$ hydrograph tests with non-overflown hydraulic barrier (*Hy-no*), with indication of the relative discharge $Q_*$ and bed load supply intensity $\Phi_i$, as a function of $t_*$.**






**Figure 15: Controlled flushing of the guiding channel after the hydrograph test *Hy-no β*, in time lapses of 0.5 $t_*$, starting from $t_* = 23.5$, after creating gradually an artificial depression above the guiding channel, until $t_* = 26.0$, where the guiding channel was completely cleared.**




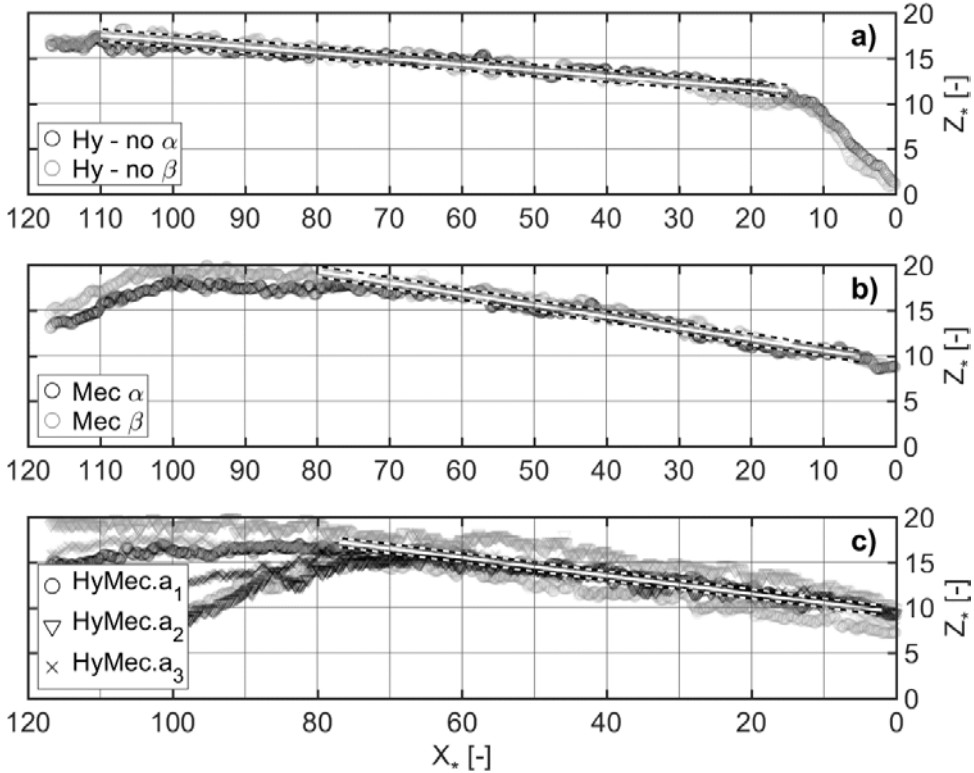

**Figure 16: Relative deposit height $Z_*$ at the longitudinal axis of the guiding channel ($Y_* = 0$) after the repetitive $\alpha$ and $\beta$ hydrograph tests; upstream of a) the non-overflown hydraulic barrier (*Hy-no*); b) the mechanical barrier only (*Mec*); and c) the combined barrier (*HyMec*) with varying opening heights $a_{1,2,3}$. The linear regression curves of the aggradation zones are shown (close white lines), with indication of the corresponding 68 % confidence intervals (dashed lines).**



**Table 1: Denomination and characterization of test runs with hydrograph and flushing episodes.**

| CASE | TYPE | REL. BARRIER HEIGHT [-] $0.11/D_{84}$ | RELATIVE HEIGHT [-] $a/D_{84}$ | CONSTRICTION WIDTH [-] $b/D_{84}$ | BAR SCREEN PLACED | HYDRO-GRAPH TESTS [N°] | FLUSH-ING |
|---|---|---|---|---|---|---|---|
| *Hy-no* | Hydraulic | *Inf.* | 11.1 | 5.6 | No | 2 | Yes |
| *Hy-o* | Hydraulic | 8.0 | $a_1 = 2.89$ | 11.0 | No | 1 | No |
| *Mec* | Mechanical | 8.0 | -- | -- | Yes | 2 | No |
| *HyMec.a₁* | Combined | 8.0 | $a_1 = 2.89$ | 11.0 | Yes | 2 | No |
| *HyMec.a₂* | Combined | 8.0 | $a_2 = 3.14$ | 11.0 | Yes | 2 | Yes |
| *HyMec.a₃* | Combined | 8.0 | $a_3 = 3.44$ | 11.0 | Yes | 2 | No |