# Peer review of "Sediment traps with guiding channel and hybrid check dams improve controlled sediment retention"

_Natural Hazards and Earth System Sciences, 2017_

## Referee Comment (RC1) · F. Comiti (Referee) · 14 Dec 2017

Dear Authors, I think your experiments were very well conducted, the results are clearly explained, and your ms is nicely written. However, I believe some clarifications are necessary before publication. Please find below my comments.

Best wishes Francesco Comiti

Introduction This section is too much focused on bedload transport issues, whereas little is presented about why check dams are used and how their implementation evolved over the past decades/centuries (see paper by Piton et al. ESPL). As the topic is very technical on check dam designing, I think the introduction should summarize the historical evolution of such structures

[Figure]

Interactive
comment

Design approach for permeable sediment traps The statement about the dominance of supercritical flows in mountain rivers is not true, as many lab and field investigations - also related to high flows - found out that critical flow conditions (Fr around 1) represent a sort of upper limit in mobile beds , for example see Grant (1997, WRR), Comiti et al. (2007, J. Hydrol, 2009, WRR), Yochum et al. (2012 J. Hydrol), Zimmermann (2012 WRR). Therefore I think you should modify your phrase, making explicit that only in the case of a smooth, stable bed (bedrock, artificial revetment) supercritical flow can onset in steep channels and thus a stable hydraulic jump can form in the retention basin downstream. Also, the assumption that bankfull discharge corresponds to effective discharge for sediment transport does not hold for steep channels (see Lenzi et al., 2006 J. Hydrol.) I would merge section 3 with section 2, as they are quite short

Methodology This section is quite well written and complete. However, I'd suggest some minor changes/comments: - the use of the term "torrential barrier": I would avoid the adjective torrential, it is not needed and in English it refers more to debris flow processes. Actually, you should clarify also earlier in the ms that debris flows are not considered in your work. - driftwood: In English refers to wood drifting in lakes or ocean, not in rivers. I'd suggest use simply "large wood" or "wood material" - Why did you choose a value of roughness n equal to 0.02 ? Please comment on its appropriateness relative to prototypes

Results Also this section reads well and presents useful information. However, I find the number of figures a bit too high and I suggest to consider removing 2-3 figures to make the paper more concise and shorter.

Discussion Can you offer an explanation for the lack of incision and reshaping of the deposit, differently from previous studies ? More in detail, is it possibly due to the relative size of sediments (with respect to flow discharges ?) Didn't previous studies obtain grain imbrication too ? Could there be a the role of test durations ? Please comment on how this lack of reworking compares to real cases You argue that the guiding channel should be rough to favor fish passage, this is correct, but isn't this

in contradiction with the Manning n=0.02 you tested ? Also, the rougher the channel the less the flushing is effective. As to driftwood passage (but please call it large wood), this can be favored for frequent, low floods and for moderate log lengths, and thus relatively large bottom openings are good also to this respect . Wood should be trapped during large, infrequent events only, as for "excessive" bedload (Comiti et al., 2016, Geomorph) The term torrential hazards again suggests debris flow-like processes in English, whereas here you mean intense bedload. I suggest to drop the term torrential

Conclusions Although your experiments do provide very interesting insights on the deposition processes during a flood, I am left with a doubt: are we sure that the guiding channels are actually beneficial for bedload permeability in the long run ? You state that after the deposition the receding flows were not able to rework the sediment deposit as the channel "attracted" the flow, leaving the deposit untouched, and then one has to intervene mechanically (with very high costs !) For ordinary floods, in a check dam without a guiding channel but with large openings the flushing could be similarly effective, I suspect. I have seen "very open" check dams which do not trap much bedload during ordinary floods, and very likely they are able to partially self-clean after a flood event through "wandering flows" over the deposit (if openings are located at different heights), apparently better than with a guiding channel (based on your experiments). The question is about how much sediment can be let pass during a flood, and this is very site specific depending on the conveyance of the downstream channel. Can you please try to "convince" more the reader on the real advantages of guiding channels ? Also, I think a big issue that you should highlight again in the conclusions is the very critical role of wood on clogging the openings, and how this should be contrasted (as discussed in the literature you already cite) or accounted for.

---

## Referee Comment (RC2) · Anonymous Referee #2 · 15 Dec 2017

**RE:** NHESS 2017 376          Schwindt et al.    Experimental study of sediments traps permeable for frequent floods

**Overview**

The authors introduce a new concept for the design of open check dam (the introduction of a guiding channel) and study through experiments its interaction with different sediments retaining techniques.

The work presented by the authors is very interesting but needs some corrections and additions before its publications.

The following are the detailed comments and specifications.

**Introduction**

Last sentence too long and confuse: please rewrite it.

**Design approach for permeable sediments trap**

The orientation of figure 1 shows a channel with an adverse slope. Could be it possible a figure with the channel inclined along the flow direction? Moreover, authors should introduce an insert or a new figure that explains the possible cases of open barriers: simple openings, bar screen or a combination of them. In present figure only the bar screen is visible.

**Experimental set up**

The Microsoft Kinect V2 seems and adapter rather than a motion-sensing camera

**Parameters and dimensional considerations**

About dimensional analysis the writer has some concern about the resulting dimensionless quantities. The dimensionless quantities should have at least one of the chosen fundamental variables. Authors should justify the presence of dimensionless quantities without them.

**Evolution bed load transfer through the barrier**

About Figure 8, could be it possible to add the inflowing sediment rate $\Phi_i$?

**Applications and limits**

The sentence at lines 17-21 of page 16 should rewritten clearly. For istance, "(steps 8 and 9 in Piton and Recking, 2016a)" should be inserted after "is similar to......." .

Bed load intensity in Figure 7 is time variant while in Figure 14 is constant.

Finally, I would suggest the authors to give just some more detail on sediment flushing and its consequence on the downstream area.

---

## Referee Comment (RC3) · Anonymous Referee #3 · 15 Dec 2017

Dear Editor, Dear Authors, I carefully reviewed the manuscript titled "Experimental study of sediment traps permeable for frequent floods" submitted as a discussion paper to the NHESS journal by Sebastian Schwindt and co-authors. I read also the other comments which have been posted. In my review I'll try, as far as possible, to avoid redundant suggestions.

General comments:

The study fits into the specific scopes of the journal since it's a potentially valuable contribution to the design, and anticipated critical evaluation of mitigation measures to reduce the impact of hazardous natural events on human-made structures and infrastructure, thereby trying to maintain or reestablish minimal levels of hydro-morphological end ecological functioning in mountain streams. In my view the

manuscript needs to be enhanced in certain aspects to reach its full potential and to be finally considered for publication in NHESS. First and crucially, the authors should better clarify the new contents with respect to the previous publication by Schwindt et al. titled "Analysis of mechanical-hydraulic bedload deposition control measures" published in Geomorphology in 2017. It's very important to minimize the overlaps and to focus almost exclusively on the analyzed new concepts for permeable sediment traps. I'm my opinion it would be advisable to summarize these previous findings in a short section titled "Current state of the experimental research" to pave the way for the presentation of the completely new research and the associated results. As stated in the abstract the new elements consist in a guiding channel featuring a permeable barrier on the downstream end. This concept is presented as completely new. I'm aware of at least two partial efforts to address a similar problem setting. One is an already implemented mitigation measure in the River Rienz in South Tyrol (compare Guis et al. 2016) and the other is an experimental study of the deposition basin in the Gadria stream as well in South Tyrol (compare https://www.baunat.boku.ac.at/fileadmin/data/H03000/H87000/H87100/IAN_Reports/REP0144.pdf). With respect to the former a sort of guiding channel has been implemented upstream of the filter (although not featuring a regular cross section). With respect to the latter the experimental variant 5 embodies as well the idea of facilitating the throughput for frequent but less intense flood events). Perhaps it could be advisable to acknowledge the existence of such partial efforts and to point out that in this study explicitly focusses on a full conceptualization. Second, it would be recommendable to extend in the introduction the description of the importance for design to quantify the nexus between an enhanced sediment flux control, reduced risks for the built environment and hydro-morphological amelioration of downstream river reaches. Additionally, I suggest to unveil the underlying design problem explicitly. Which real world problem are you attempting to solve? Sparsely throughout the text you report field data, so it would be interesting to know if a real world case (or more than one) motivated your study. This is not of minor importance, since an optimal functioning of a certain sediment dosing

or filtering system can be ultimately judged based on the sediment supply needs of the specific river and the natural hazard risk of the specific built environment. Has a specific design objective been defined in terms of risk reduction, eco-morphological enhancement, cost minimization?

Specific comments:

Title: As a result of the revision process the authors should judge if the title merits to be slightly adapted.

Introduction: 1) Is the effective or dominant discharge also a useful concept in heavily modified (e.g. by check dams) alpine mountain torrents. If not, it would be interesting to know how the sediment demand for downstream reaches should be assessed. In my view this is a crucial design element. 2) The references are sometimes presented in chronologically ascending order and sometimes not, please adhere to the journal guidelines in this respect. 3) You mention that "The application of the grain size of the traveling bed load to bed load transport formulae can be used for establishing sediment rating curves, as a computation basis for the dominant discharge." Which grain sixe exactly? Please specify. 4) You use the terms eco-morphological depletion. I prefer degradation. Consider revising your wording.

Design approach for permeable sediment traps

If possible, the design approach should be presented in a much more coherent way. For example, stating, first, the objective(s) of the design and the applicable design principles and, consequently, the physical effects to be achieved and, lastly, the detailed structural design.

Methodology – Experimental setup: You start by "The design of the experimental set-up (Figure 3) was inspired by 132 characteristic datasets from mountain rivers (Schwindt, 2017). Thus, even though any particular prototype underlay the model, a geometric scale in the range of 1:10 to 1:40 can be supposed." I think it would be advantageous

to specify what exactly inspired the design of the experimental set-up. Moreover, also the second sentence needs further clarification.

Subsection 4.2 Deposition area with guiding channel: I'm not particularly convinced of the effectiveness of this subsection title.

Further minor issues:

Section 5 – Results and Analysis. Also in this case I urge the authors to slightly change the section title. Check carefully the reference style to consistently use brackets where needed. Moreover, equations should not contain references (e.g., Johnson, 2016).
* * *

---

## Author Comment (AC1) · 15 Jan 2018

**Author's response**

**F. Comiti (Referee)**

francesco.comiti@unibz.it

Dear Authors,

I think your experiments were very well conducted, the results are clearly explained, and your ms is nicely written. However, I believe some clarifications are necessary before publication. Please find below my comments.

Best wishes Francesco Comiti

*Dear Francesco Comiti,*

*Thank you for your thorough and constructive review of our Manuscript. We adapted the text in response to your comments. We answer the particular remarks in detail below. The updated manuscript still requires the Editor's invitation that we hope to receive soon.*

*Best wishes,*

*the Authors.*

**Introduction**

This section is too much focused on bedload transport issues, whereas little is presented about why check dams are used and how their implementation evolved over the past decades/centuries (see paper by Piton et al. ESPL). As the topic is very technical on check dam designing, I think the introduction should summarize the historical evolution of such structures

*We improved the introduction by completing the reasoning for the construction of check dams, also applying on Piton et al. (2017).*

**Design approach for permeable sediment traps**

The statement about the dominance of supercritical flows in mountain rivers is not true, as many lab and field investigations - also related to high flows - found out that critical flow conditions (Fr around 1) represent a sort of upper limit in mobile beds , for example see Grant (1997, WRR), Comiti et al. (2007, J. Hydrol, 2009, WRR), Yochum et al. (2012 J. Hydrol), Zimmermann (2012 WRR). Therefore I think you should modify your phrase, making explicit that only in the case of a smooth, stable bed (bedrock, artificial revetment) supercritical flow can onset in steep channels and thus a stable hydraulic jump can form in the retention basin downstream.

*We adapted the text.*

Also, the assumption that bankfull discharge corresponds to effective discharge for sediment transport

does not hold for steep channels (see Lenzi et al., 2006 J. Hydrol.)

*We completely agree with that and we clarify this differentiation now in the revised discussion Section 5.4 (Application and limits).*

I would merge section 3 with section 2, as they are quite short

*This is true. We merged sections 2 and 3 but we kept subsections because both titles are two different aspects that are crucial for the understanding of the paper. We want the reader to easily relocate both the design of check dams and the related sediment retention pattern through the section titles.*

**Methodology**

This section is quite well written and complete. However, I'd suggest some minor changes/comments:
- the use of the term "torrential barrier": I would avoid the adjective torrential, it is not needed and in English it refers more to debris flow processes.

*We removed the adjective "torrential" in front of "barrier" in the manuscript.*

Actually, you should clarify also earlier in the ms that debris flows are not considered in your work.

*We added this hint at the end of the introduction.*

- driftwood: In English refers to wood drifting in lakes or ocean, not in rivers. I'd suggest use simply "large wood" or "wood material" –

*Implemented.*

Why did you choose a value of roughness n equal to 0.02 ? Please comment on its appropriateness relative to prototypes

*The roughness originates from the bed grain size and it results from respecting the geometry scales that we observed in nature (e.g., ratio between channel width and grain size). We assessed the interpolated Manning's n in earlier studies using a shooting algorithm applied to the resolution of 1D Saint-Venant equations along the channel. We added this in the text.*

**Results**

Also this section reads well and presents useful information. However, I find the number of figures a bit too high and I suggest to consider removing 2-3 figures to make the paper more concise and shorter.

*The application of the motion-sensing device (Kinect V2) is interesting to know but not crucial for our manuscript. Therefore, we provide the former Figure 9 (Kinect application) and the error evaluation (Figure 10) as supplemental material now.*

**Discussion**

Can you offer an explanation for the lack of incision and reshaping of the deposit, differently from previous studies? More in detail, is it possibly due to the relative size of sediments (with respect to flow discharges?)

*Indeed, we used a larger grain mixture than what was used in previous studies and we added this hint in the text. However, given the small geometric scale (large size) of our model, the grain sizes are coherent with the set of field observations that we used.*

Didn't previous studies obtain grain imbrication too?

*This would be interesting to know but we cannot appropriately judge imbrication based on the reports from former studies.*

Could there be a the role of test durations?

*Sure, there is a role of duration. We tested longer durations in preliminary tests and we observed more pronounced sediment deposition and higher deposit volumes in these tests. Therefore, our*

*experiments are on the safe site regarding sediment retention. We added this in the discussion.*

Please comment on how this lack of reworking compares to real cases You argue that the guiding channel should be rough to favor fish passage, this is correct, but isn't this in contradiction with the Manning n=0.02 you tested? Also, the rougher the channel the less the flushing is effective.

*The situation that we modeled corresponds to floods when, we assume, there is no fish migration. In the flood situation, the relative grain submergence is low, and therefore the flow smoothens. We added this aspect to the discussion of eco-morphological aspects in Section 5.3.*

As to driftwood passage (but please call it large wood), this can be favored for frequent, low floods and for moderate log lengths, and thus relatively large bottom openings are good also to this respect. Wood should be trapped during large, infrequent events only, as for "excessive" bedload (Comiti et al., 2016, Geomorph)

*We added this and the literature resource in the Manuscript.*

The term torrential hazards again suggests debris flow-like processes in English, whereas here you mean intense bedload. I suggest to drop the term torrential

*As earlier proposed, we dropped the term torrential with respect to our study but in the reference to Piton and Recking (2016a), we kept the term, as it applies to their study.*

**Conclusions**

Although your experiments do provide very interesting insights on the de- position processes during a flood, I am left with a doubt: are we sure that the guiding channels are actually beneficial for bedload permeability in the long run? You state that after the deposition the receding flows were not able to rework the sediment deposit as the channel "attracted" the flow, leaving the deposit untouched, and then one has to intervene mechanically (with very high costs !) For ordinary floods, in a check dam without a guiding channel but with large openings the flushing could be similarly effective, I suspect.

*The key point is that the sediment transfer is improved through the guiding channel up to small, non-hazardous floods. Mechanical interventions after an important flood event are inevitable. We adapted the conclusions to highlight this important aspect.*

I have seen "very open" check dams which do not trap much bedload during ordinary floods, and very likely they are able to partially self-clean after a flood event through "wandering flows" over the deposit (if openings are located at different heights), apparently better than with a guiding channel (based on your experiments). The question is about how much sediment can be let pass during a flood, and this is very site specific depending on the conveyance of the downstream channel. Can you please try to "convince" more the reader on the real advantages of guiding channels?

*The self-cleaning may be interesting but it can also be very dangerous. We added a paragraph in the discussion Section 5.2 (sediment flushing) to underline that (also according to a remark from Reviewer 3). With our sediment trap concept, we want to promote sediment continuity but only if the risk of uncontrolled self-cleaning can be avoided. We added the key word of self cleaning and we adapted the conclusions to stress the importance of avoiding unwanted sediment flushing.*

Also, I think a big issue that you should highlight again in the conclusions is the very critical role of wood on clogging the openings, and how this should be contrasted (as discussed in the literature you already cite) or accounted for.

*We added this for future work at the end of the conclusions.*

---

## Author Comment (AC2) · 15 Jan 2018

**RE:** *NHESS 2017 376   Schwindt et al.: Experimental study of sediments traps permeable for frequent floods*

*Authors response*

**Overview**

The authors introduce a new concept for the design of open check dam (the introduction of a guiding channel) and study through experiments its interaction with different sediments retaining techniques.

The work presented by the authors is very interesting but needs some corrections and additions before its publications.

*Dear Reviewer,*

*Thank you for the time you took for reviewing our Manuscript. We appreciate your comments and we applied them to our manuscript according to the detailed responses. The updated manuscript still requires the Editor's invitation that we hope to receive soon.*

*Kind regards,*

*the Authors*

The following are the detailed comments and specifications.

**Introduction**

Last sentence too long and confuse: please rewrite it.

*We improved the introduction, also accounting for the comments from Francesco Comiti (RC 1) and Reviewer 3.*

**Design approach for permeable sediments trap**

The orientation of figure 1 shows a channel with an adverse slope. Could be it possible a figure with the channel inclined along the flow direction?

*Our main study objectives address the deposition area and the barrier (check dam) which are only visible from an upstream point of view. We created Figure 1 based on a CAD drawing where the channel has a geometric slope in the flow direction. If we had changed the view angle, it was not possible to show the target elements of our study. Thus, the CAD model that constitutes Figure 1 is correct regarding the landscape and structure geometry. We hope that you agree that we kept Figure 1 as it is, because changes in the view angle would reduce the comprehension of the elements in our study.*

Moreover, authors should introduce an insert or a new figure that explains the possible cases of open barriers: simple openings, bar screen or a combination of them. In present figure only the bar screen is visible.

*A number of standard literature discusses and describes the barrier types and their openings. Any reproduction of these figures or only similar representations in a journal article require copyright authorizations. Even though we agree that an overview on existing barriers can be interesting for the reader but it is not crucial for understanding our study. We added a paragraph that cites sources of*

*check dam design charts; this literature includes the phd thesis of the main author, which is publicly available (we added the public access url in the list of references: Schwindt, 2017).*

**Experimental set up**

The Microsoft Kinect V2 seems and adapter rather than a motion-sensing camera

*The manual mentions the device as "consumer-grade RGB-D sensor", i.e., neither camera nor adapter. We changed the term "camera" to "device" to avoid confusion.*

**Parameters and dimensional considerations**

About dimensional analysis the writer has some concern about the resulting dimensionless quantities. The dimensionless quantities should have at least one of the chosen fundamental variables. Authors should justify the presence of dimensionless quantities without them.

*We choose this set of fundamental variables that suits our analysis of sediment transport-related phenomena, as we consider bed load transport as the principal process in our study. We use ratios of other base variables, e.g., for hydrodynamic parameters, where our set of fundamental variables is not accurate (Yalin, 1977). These parameters are, e.g., the discharge where we use the bankfull discharge of the guiding channel for normalization, or the duration of the hydrograph. Prior to our analysis, we also considered a normalization of discharge or time based on the fundamental variables but such representations are not meaningful for the interpretation of the results. If we were applying the fundamental variables to the time, then $t^* = t/sqrt(g \cdot D_{84})$, i.e., a variable that has no meaning for the analysis. Nevertheless, all geometry-related dimensionless parameters result from the fundamental variable of the $D_{84}$, where the set of fundamental parameters corresponds to the one used by Einstein (1950) to derive $\Phi_i$. Further discussions on the set of fundamental variables and dimensional analysis are included in the main authors phd thesis (Schwindt, 2017) and in the former articles that originate from previous adjustments of the same experimental setup (Schwindt et al. 2017a, 2017b, 2017c). However, we cannot find an accurate place for these explanations in the dimensional analysis-section without dispersing the reader from the main objective of our in the journal of Natural Hazards and Earth System Sciences.*

**Evolution bed load transfer through the barrier**

About Figure 8, could be it possible to add the inflowing sediment rate $\Phi_i$?

*We considered adding the sediment supply rate but the graphs became very messy and hard to read – here an example of Fig. 8a, where the graph density is still much lower than in Fig. 8c:*

[Figure]

*In particular, the range, where the bed load outflow and inflow rates are present, is very hard to interpret. This is why we opted to present the sediment supply rate apart from Fig. 8, with the hydrograph in Fig. 7.*

**Applications and limits**

The sentence at lines 17-21 of page 16 should rewritten clearly. For istance, "(steps 8 and 9 in Piton and Recking, 2016a)" should be inserted after "is similar to……." .

*We adapted the text as proposed (with the new numbering, the paragraph is in Section 5.4).*

Bed load intensity in Figure 7 is time variant while in Figure 14 is constant.

*Figure 7 shows the hydrograph test while Figure 14 shows the flushing test. We added information on the constant sediment supply at the beginning of the flushing experiments in Section 3.4 (Experimental procedures).*

Finally, I would suggest the authors to give just some more detail on sediment flushing and its consequence on the downstream area.

*We added a comment on the consequences of sediment flushing for downstream reaches at the end of the discussion section 5.2 (Sediment flushing).*

---

## Author Comment (AC3) · 15 Jan 2018

**Author's response**

**Anonymous Referee #3**

Dear Editor,

Dear Authors,

I carefully reviewed the manuscript titled "Experimental study of sediment traps permeable for frequent floods" submitted as a discussion paper to the NHESS journal by Sebastian Schwindt and co-authors. I read also the other comments which have been posted. In my review I'll try, as far as possible, to avoid redundant suggestions.

*Dear Reviewer,*

*Thank you for the ample review and the constructive suggestions for improving our Manuscript. We answer your General and Specific comments individually. We hope that our reviewed Manuscript satisfies your concerns. The updated manuscript still requires the Editor's invitation that we hope to receive soon.*

*Kind regards,*

*the Authors*

**General comments:**

The study fits into the specific scopes of the journal since it's a potentially valuable contribution to the design, and anticipated critical evaluation of mitigation measures to reduce the impact of hazardous natural events on human-made structures and infrastructure, thereby trying to maintain or reestablish minimal levels of hydro- morphological end ecological functioning in mountain streams. In my view the manuscript needs to be enhanced in certain aspects to reach its full potential and to be finally considered for publication in NHESS. First and crucially, the authors should better clarify the new contents with respect to the previous publication by Schwindt et al. titled "Analysis of mechanical-hydraulic bedload deposition control measures" published in Geomorphology in 2017. It's very important to minimize the overlaps and to focus almost exclusively on the analyzed new concepts for permeable sediment traps. I'm my opinion it would be advisable to summarize these previous findings in a short section titled "Current state of the experimental research" to pave the way for the presentation of the completely new research and the associated results.

*We adapted the introduction to make clear that the previous study introduced the hybrid control barrier in a flume only and we emphasize more the framework of the new experiments in this study.*

As stated in the abstract the new elements consist in a guiding channel featuring a permeable barrier on the downstream end. This concept is presented as completely new. I'm aware of at least two partial efforts to address a similar problem setting. One is an already implemented mitigation measure in the River Rienz in South Tyrol (compare Guis et al. 2016) and the other is an experimental study of the deposition basin in the Gadria stream as well in South Tyrol (compare

https://www.baunat.boku.ac.at/fileadmin/data/H03000/H87000/H87100/IAN_Reports/REP0144.pdf).
With respect to the former a sort of guiding channel has been implemented upstream of the filter (although not featuring a regular cross section). With respect to the latter the experimental variant 5 embodies as well the idea of facilitating the throughput for frequent but less intense flood events). Perhaps it could be advisable to acknowledge the existence of such partial efforts and to point out that in this study explicitly focusses on a full conceptualization.

*We added comments on similar features in the text and we added the report from Hübl et al.. Alas, we could neither retrieve Guis et al. (2016) nor any sediment trap/check dam/guiding channel related study at the Rienz River.*

Second, it would be recommendable to extend in the introduction the description of the importance for design to quantify the nexus between an enhanced sediment flux control, reduced risks for the built environment and hydro-morphological amelioration of downstream river reaches.

*We adapted the introduction, also according to the comments from Francesco Comiti (RC 1) and the Specific comments.*

Additionally, I suggest to unveil the underlying design problem explicitly. Which real world problem are you attempting to solve? Sparsely throughout the text you report field data, so it would be interesting to know if a real world case (or more than one) motivated your study. This is not of minor importance, since an optimal functioning of a certain sediment dosing or filtering system can be ultimately judged based on the sediment supply needs of the specific river and the natural hazard risk of the specific built environment. Has a specific design objective been defined in terms of risk reduction, eco-morphological enhancement, cost minimization?

*We made major adaptations in the introduction in general and the introduction of our concept. Furthermore, we enhanced the discussion section regarding the application (Section 5.4) for better highlighting the context, purpose and utility of our tested concept.*

**Specific comments:**

Title: As a result of the revision process the authors should judge if the title merits to be slightly adapted.

*We adapted the title to "Sediment traps with guiding channel and hybrid check dams improve controlled sediment retention" (pending upload invitation for the reviewed manuscript).*

**Introduction**:

1) Is the effective or dominant discharge also a useful concept in heavily modified (e.g. by check dams) alpine mountain torrents. If not, it would be interesting to know how the sediment demand for downstream reaches should be assessed. In my view this is a crucial design element.

*We introduce here aspects on sediment transport and its assessment in general. We enhanced the discussion on the application of our tested concept (Section 5.4).*

2) The references are sometimes presented in chronologically ascending order and sometimes not, please adhere to the journal guidelines in this respect.

*We generally adapted the citations to the increasing-order style.*

3) You mention that "The application of the grain size of the traveling bed load to bed load transport formulae can be used for establishing sediment rating curves, as a computation basis for the dominant discharge." Which grain sixe exactly? Please specify.

*The recent scientific literature refers to the $D_{84}$ for the assessment of roughness while the mean grain size of overbank sediment deposits provides accurate estimates for the sediment flux. We adapted the introduction accordingly.*

4) You use the terms eco-morphological depletion. I prefer degradation. Consider revising your wording.

*We replaced depletion by degradation.*

**Design approach for permeable sediment traps**

If possible, the design approach should be presented in a much more coherent way. For example, stating, first, the objective(s) of the design and the applicable design principles and, consequently, the physical effects to be achieved and, lastly, the detailed structural design.

*We rearranged this section according to the comment and we deleted repetitive information.*

**Section 4 – Methodology**

Experimental setup: You start by "The design of the experimental set-up (Figure 3) was inspired by 132 characteristic datasets from mountain rivers (Schwindt, 2017). Thus, even though any particular prototype underlay the model, a geometric scale in the range of 1:10 to 1:40 can be supposed." I think it would be advantageous to specify what exactly inspired the design of the experimental set-up. Moreover, also the second sentence needs further clarification.

*We were interested in typical geometric relationships (grain size, channel width, flow depth) and discharges. We added this hint in the text.*

Subsection 4.2 Deposition area with guiding channel: I'm not particularly convinced of the effectiveness of this subsection title.

*We changed the subsections' title to "Premises and descriptions of the deposition area with guiding channel".*

Further minor issues:

**Section 5 – Results and Analysis**. Also in this case I urge the authors to slightly change the section title. Check carefully the reference style to consistently use brackets where needed. Moreover, equations should not contain references (e.g., Johnson, 2016).

*We corrected the title of subsection 4.1 (former 5.1): "Evolution of bed load transfer through the barrier". In addition, we changed the subscript in Equation (3) from citation style to a normal subscript that still is unique and makes the origin of this equation clear. Moreover, we updated the references.*

*We made further changes according to the comments from Francesco Comiti (RC 1) and Reviewer 2.*